# Comprehensive fitness maps of Hsp90 show widespread environmental dependence

**Julia M Flynn[1], Ammeret Rossouw[1], Pamela Cote-Hammarlof[1], Inês Fragata[2], David Mavor[1], Carl Hollins III[1], Claudia Bank[2], Daniel NA Bolon[1]***

[1]Department of Biochemistry and Molecular Pharmacology, University of Massachusetts Medical School, Worcester, United States; [2]Instituto Gulbenkian de Ciência, Oeiras, Portugal

**Abstract** Gene-environment interactions have long been theorized to influence molecular evolution. However, the environmental dependence of most mutations remains unknown. Using deep mutational scanning, we engineered yeast with all 44,604 single codon changes encoding 14,160 amino acid variants in Hsp90 and quantified growth effects under standard conditions and under five stress conditions. To our knowledge, these are the largest determined comprehensive fitness maps of point mutants. The growth of many variants differed between conditions, indicating that environment can have a large impact on Hsp90 evolution. Multiple variants provided growth advantages under individual conditions; however, these variants tended to exhibit growth defects in other environments. The diversity of Hsp90 sequences observed in extant eukaryotes preferentially contains variants that supported robust growth under all tested conditions. Rather than favoring substitutions in individual conditions, the long-term selective pressure on Hsp90 may have been that of fluctuating environments, leading to robustness under a variety of conditions.

## Introduction

The role of environment has been contemplated in theories of evolution for over a hundred years (*Darwin, 1859*; *Darwin and Wallace, 1858*; *Wright, 1932*), yet molecular level analyses of how environment impacts the evolution of gene sequences remain experimentally under-explored. Depending on environmental conditions, mutations can be categorized into three classes: strongly deleterious mutations that are purged from populations by purifying selection, nearly-neutral mutations that are governed by stochastic processes, and beneficial mutations that tend to provide a selective advantage (*Ohta, 1973*). It has long been clear that environmental conditions can alter the fitness effects of mutations (*Tutt, 1896*). However, examining how environmental conditions impact any of the three classes of mutations is challenging. Measurable properties of nearly-neutral and deleterious mutations in natural populations are impacted by both demography and selection (*Ohta, 1973*), which are difficult to disentangle. In addition, many traits are complex, making it challenging to identify all contributing genetic variations (*McCarthy et al., 2008*). For these and other reasons, we do not have a detailed understanding of how environmental conditions impact the evolution of most gene sequences.

Mutational scanning approaches (*Fowler et al., 2010*) provide novel opportunities to examine fitness effects of the same mutations under different laboratory conditions (*Boucher et al., 2016*; *Boucher et al., 2014*; *Canale et al., 2018*; *Kemble et al., 2019*). The EMPIRIC (Exceedingly Meticulous and Parallel Investigation of Randomized Individual Codons) approach that we previously developed (*Hietpas et al., 2011*) is particularly well suited to address questions regarding the environmental impact of mutational effects for three reasons: it quantifies growth rates that are a

***For correspondence:**
Dan.Bolon@umassmed.edu

**Competing interests:** The authors declare that no competing interests exist.

direct measure of experimental fitness, all point mutations are engineered providing comprehensive maps of growth effects, and all the variants can be tracked in the same flask while experiencing identical growth conditions. We have previously used the EMPIRIC approach to investigate how protein fitness maps of ubiquitin vary in different environmental conditions (*Mavor et al., 2016*). The analysis of ubiquitin fitness maps revealed that stress environments can exacerbate the fitness defects of mutations. However, the small size of ubiquitin and the near absence of natural variation in ubiquitin sequences (only three amino acid differences between yeast and human) hindered investigation of the properties underlying historically observed substitutions.

Mutational scanning approaches have emerged as a robust method to analyze relationships between gene sequence and function, including aspects of environmental-dependent selection pressure. Multiple studies have investigated resistance mutations that enhance growth in drug or antibody environments (*Dingens et al., 2019*; *Doud et al., 2018*; *Firnberg et al., 2014*; *Jiang et al., 2016*; *Stiffler et al., 2015*). Most of these studies have focused on interpreting adaptation in the light of protein structure. Of note, Dandage, Chakraborty and colleagues explored how environmental perturbations to protein folding influenced tolerance of mutations in the 178 amino acid gentamicin-resistant gene in bacteria (*Dandage et al., 2018*). However, the question of how environmental variation shapes the selection pressure on gene sequences has not been well studied.

Here, we report comprehensive experimental fitness maps of Heat Shock Protein 90 (Hsp90) under multiple stress conditions and compare our experimental results with the historical record of hundreds of Hsp90 substitutions accrued during its billion years of evolution in eukaryotes. Hsp90 encodes a 709 amino acid protein and to our knowledge it is the largest gene for which a comprehensive protein fitness map has been determined. Hsp90 is an essential and highly abundant molecular chaperone which is induced by a wide variety of environmental stresses (*Gasch et al., 2000*; *Lindquist, 1981*). Hsp90 assists cells in responding to these stressful conditions by facilitating the folding and activation of client proteins through a series of ATP-dependent conformational changes mediated by co-chaperones (*Krukenberg et al., 2011*). These clients are primarily signal transduction proteins, highly enriched in kinases and transcription factors (*Taipale et al., 2012*). Through its clients, Hsp90 activity is linked to virtually every cellular process.

Hsp90 can facilitate the emergence and evolution of new traits in response to stress conditions, including drug resistance in fungi (*Cowen and Lindquist, 2005*), gross morphology in flies (*Rutherford and Lindquist, 1998*) and plants (*Queitsch et al., 2002*), and vision loss in cave fish (*Rohner et al., 2013*). In non-stress conditions, an abundance of Hsp90 promotes standing variation by masking the phenotypic effects of destabilizing mutations in clients. Stressful conditions that tax Hsp90 capacity can then manifest in phenotypic diversity that can contribute to adaptation. Because of the biochemical and evolutionary links between Hsp90 and stress, we hypothesized that environmental stress would result in altered fitness maps.

The conditions in natural environments often fluctuate, and all organisms contain stress response systems that aid in acclimation to new conditions. The conditions experienced by different populations can vary tremendously depending on the niches that they inhabit, providing the potential for distinct selective pressures on Hsp90. Previous studies of a nine amino acid loop in Hsp90 identified multiple amino acid changes that increased the growth rate of yeast in elevated salinity (*Hietpas et al., 2013*), demonstrating the potential for environmental-dependent beneficial mutations in Hsp90. However, the sequence of Hsp90 is strongly conserved in eukaryotes (57% amino acid identity from yeast to human), indicating consistent strong purifying selection.

To investigate the potential influence of the environment on Hsp90 evolution, we quantified fitness maps in six different conditions. While proximity to ATP is the dominant functional constraint in standard conditions, the influence of client and co-chaperone interactions on growth rate dramatically increases under stress conditions. Increased selection pressure from heat and diamide stresses led to a greater number of beneficial variants compared to standard conditions. The observed beneficial variants were enriched at functional hotspots in Hsp90. However, the natural variants of Hsp90 tend to support efficient growth in all environments tested, indicating selection for robustness to diverse stress conditions in the natural evolution of Hsp90.

# Results

We developed a powerful experimental system to analyze the growth rate supported by all possible Hsp90 point mutations under distinct growth conditions. Bulk competitions of yeast with a deep sequencing readout enabled the simultaneous quantification of 98% of possible amino acid changes (*Figure 1A*). The single point mutant library was engineered by incorporating a single degenerate codon (NNN) into an otherwise wild-type Hsp90 sequence as previously described (*Hietpas et al., 2012*). To provide a sensitive readout of changes in Hsp90 function, we transferred the library to a plasmid under a constitutive low-expression level ADH promoter that reduced Hsp90 protein levels to near-critical levels (*Jiang et al., 2013*). To efficiently track all variants in a single competition flask so that all variants experience identical conditions, we updated our previously developed EMPIRIC approach to include a barcoding strategy (*Hietpas et al., 2012*). As described in the Materials and methods, this barcode strategy enabled us to track mutations across a large gene using a short sequencing readout.

We transformed the plasmid library of comprehensive Hsp90 point mutations into a conditional yeast strain where we could turn selection of the library on or off. We used a yeast Hsp90-shutoff strain in which both paralogs of Hsp90 (*hsc82* and *hsp82*) are deleted and a copy of *hsp82* with expression under strict regulation of a galactose-inducible promoter is integrated into the chromosome (*Jiang et al., 2013*). The mutant libraries were amplified in the absence of selection on the mutant variant by growing the transformed yeast in galactose media that expresses the wild-type chromosomal copy of *hsp82*. We switched the yeast to dextrose media to shut off the expression of wild-type Hsp90, allowing the mutagenized variants to be the sole source of Hsp90 protein in the cell, and then split the culture into six different environmental conditions. We extracted samples from each condition at multiple time points and used Illumina sequencing to estimate the frequency of each Hsp90 variant over time. We assessed the selection coefficient of each Hsp90 variant from the change in frequency relative to wild-type Hsp90 using a previously developed Bayesian Markov Chain Monte Carlo (MCMC) method (*Bank et al., 2014*; *Fragata et al., 2018*), where 0 represents wild-type and −1 represents null alleles (*Figure 1—source data 1*).

To analyze reproducibility of the growth competition, we performed a technical replicate under standard conditions. We used a batch of the same transformed cells that we had frozen and stored such that the repeat bulk competition experiments and sequencing were performed independently. Selection coefficients between replicates were strongly correlated ($R^2 = 0.90$), and indicated that we could clearly distinguish between selection coefficients for members of the library containing silent mutations that do not change the amino acid sequence (wild-type synonyms) and those containing stop codons (*Figure 1B*, *Figure 1—figure supplement 1*). For the second replicate we noted a small fitness defect (s ~ − 0.2) for wild-type synonyms at positions 679–709 relative to other positions (*Figure 1—figure supplement 1*). We did not see this behavior in any other condition or replicate tested and do not understand its source. The selection coefficients in this study under standard conditions also correlated strongly ($R^2 = 0.87$) with estimates of the Hsp90 N-domain in a previous study (*Mishra et al., 2016*; *Figure 1—figure supplement 2*), indicating that biological replicates also show high reproducibility. Of note, variants with strongly deleterious effects exhibited the greatest variation between replicates, consistent with the noise inherent in estimating the frequency of rapidly depleting variants (*Figure 1B*). The stop codons were already partially depleted from the cells at the 0 time point, likely contributing to their variation between replicates (*Figure 1—figure supplement 3A*). In accordance with this, there was a higher variation in selection coefficients between replicates for stop codons with the lowest initial reads (*Figure 1—figure supplement 3B*). Stop codon fitness was similar for all three stop codons (*Figure 1—figure supplement 3C*) and at positions across Hsp90 with exception of the last 32 positions that have previously been shown to be dispensable for its viability (*Louvion et al., 1996*; *Figure 1—figure supplement 1*). A heatmap representation of all the selection coefficients determined in standard conditions in replicate one is shown in *Figure 1—figure supplement 4*.

The large number of signaling pathways that depend on Hsp90 (*Taipale et al., 2012*) and its strong sequence conservation suggest that many mutations of Hsp90 may decrease fitness. However, most variants of Hsp90 had wild-type-like fitness in the competition experiment in standard conditions (*Figure 1C*, *Figure 1—figure supplement 4*). All possible mutations (excluding stops) were compatible with function at 425 positions. Only 17 positions had low mutational tolerance to

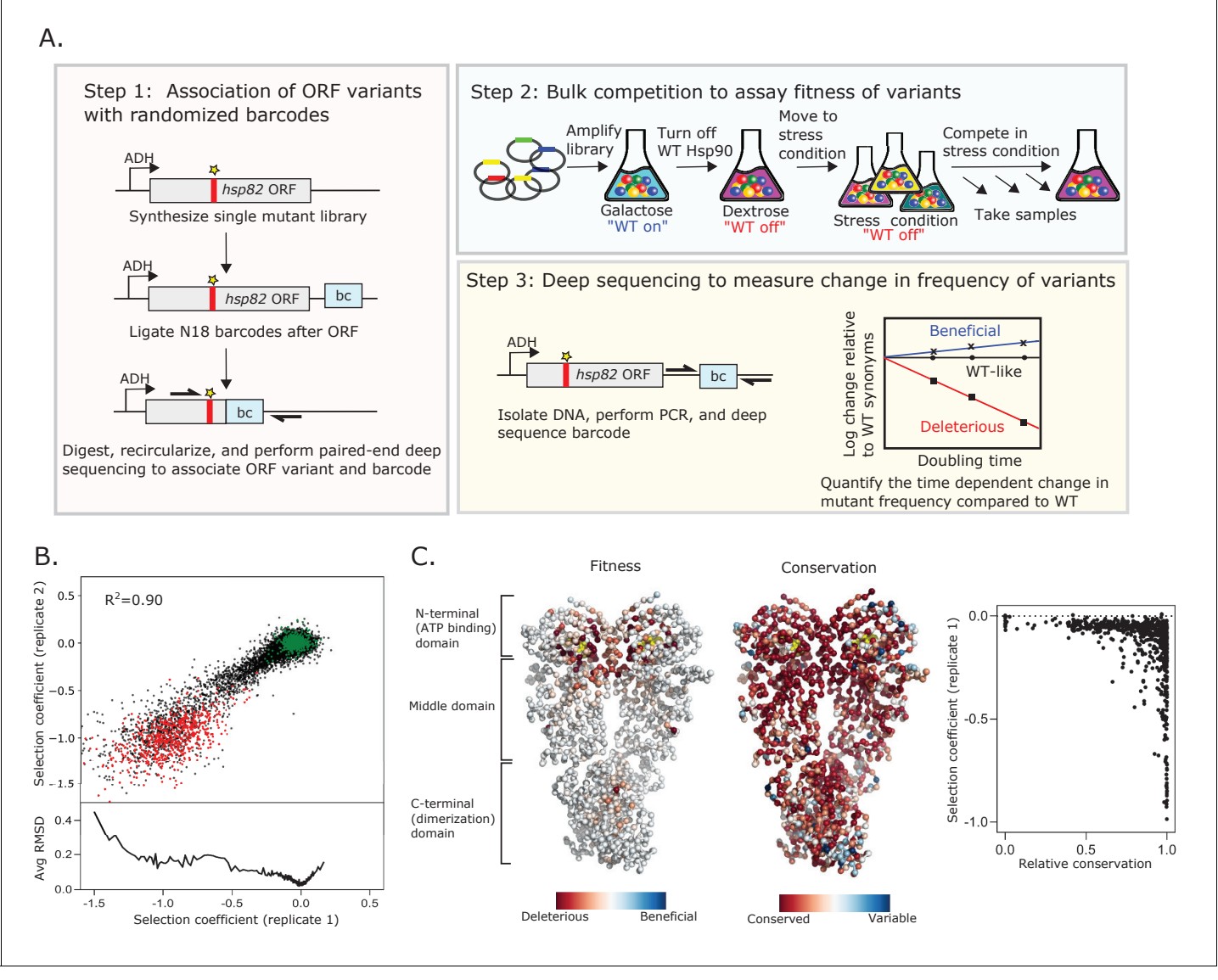

**Figure 1.** Approach to determine protein fitness maps of Hsp90. (A) Barcoded competition strategy to analyze the growth effects of all single codon variants of Hsp90 in a single bulk culture. *Hsp82* is the stress-inducible gene that encodes for Hsp90 (B) Measurements of selection coefficients of amino acid variants are reproducible in replicate growth competitions (see *Figure 1—source data 1*). Wild-type amino acids are shown in green and stop codons are shown in red. The bottom panel shows the Root-mean-square deviation (RMSD) averaged for a running window of 40 data points. (C) Average selection coefficients at each position in standard conditions mapped onto a homodimeric structure of Hsp90 (PDB 2cg9, *Ali et al., 2006*) and compared to patterns of evolutionary conservation (see *Figure 1—source data 2*). ATP is shown in yellow. The graph on the right compares relative conservation at each position of Hsp90 to the average selection coefficient at that position.

The online version of this article includes the following source data and figure supplement(s) for figure 1:

**Source data 1.** Sequencing counts and selection coefficients for each individual amino acid change across amino acids 2–709 of Hsp90 in both replicates of standard conditions.
**Source data 2.** Average selection coefficient (excluding stops) at each position of Hsp90 in Standard replicate 1.
**Figure supplement 1.** Selection coefficients for wild-type synonyms (green) and stops (red) at each position of Hsp90 for both replicates of standard conditions.
**Figure supplement 2.** Measurement of selection coefficients for positions 2–220 in this study correlated strongly ($R^2$ = 0.87) with estimates of the Hsp90 N-domain in a previous study (*Mishra et al., 2016*), indicating that biological replicates show high reproducibility.
**Figure supplement 3.** Analysis of variation in stop codon selection coefficients.
**Figure supplement 4.** Heatmap representation of the selection coefficients observed for single amino acid changes across amino acids 2–709 of Hsp90 in standard (30°C) conditions in replicate 1.
**Figure supplement 5.** Correlation of mutational sensitivity with distance to ATP.

the extent that 15 or more substitutions caused null-like growth defects (R32, E33, N37, D40, D79, G81, G94, I96, A97, S99, G118, G121, G123, Y125, F156, W300, and R380). All these positions except for W300 are in contact with ATP or mediate ATP-dependent conformational changes in the N-domain of Hsp90. In fact, the average selection coefficient at different positions (a measure of mutational sensitivity) in standard growth conditions correlates ($R^2 = 0.49$) with distance from ATP (*Figure 1—figure supplement 5*). While W300 does not contact ATP, it transmits information from client binding to long range conformational changes of Hsp90 that are driven by ATP hydrolysis (*Röhl et al., 2013*). Our results indicate that ATP binding and the conformational changes driven by ATP hydrolysis impose dominant physical constraints in Hsp90 under standard laboratory conditions.

At first sight, the observation that most mutations are compatible with robust growth in standard conditions is at odds with the fact that the Hsp90 sequence is strongly conserved across large evolutionary distances (*Figure 1C*). One potential reason for this discrepancy could be that the strength of purifying selection in large natural populations over long evolutionary time-scales is more stringent than can be measured in the laboratory. In other words, experimentally unmeasurable fitness defects could be subject to purifying selection in nature. In addition, the range of environmental conditions that yeast experience in natural settings may not be reflected by standard laboratory growth conditions. To investigate the impact of environmental conditions on mutational effects in Hsp90, we measured the growth rate of Hsp90 variants under five additional stress conditions.

## Impact of stress conditions on mutational sensitivity of Hsp90

We measured the fitness of Hsp90 variants in conditions of nitrogen depletion (ND) (0.0125% ammonium sulfate), hyper-osmotic shock (0.8 M NaCl), ethanol stress (7.5% ethanol), the sulfhydryl-oxidizing agent diamide (0.85 mM), and temperature shock (37°C). All these stresses are known to elicit a common shared environmental stress response characterized by altered expression of ~900 genes as well as having specific responses unique to each stress (*Gasch et al., 2000*). Genes encoding heat-shock proteins, including Hsp90, are transiently upregulated in all these stresses except elevated salinity (*Gasch et al., 2000*; *Piper, 1995*).

One way to characterize stress conditions is to measure the extent to which they slow down growth. For our experiments, each of the environmental stresses were selected to partially decrease the growth rate. Consistently, all stresses reduced the growth rate of the parental strain within a two-fold range, with depletion of nitrogen levels causing the smallest reduction in growth rate and diamide causing the greatest reduction (*Figure 2A*). To investigate how critical Hsp90 is for growth in each condition, we measured growth rates of yeast with either normal or more than 10-fold reduced (*Jiang et al., 2013*) levels of Hsp90 protein (*Figure 2A*). Under standard conditions, the normal level of Hsp90 protein can be dramatically reduced without major impacts on growth rate, consistent with previous findings (*Jiang et al., 2013*; *Picard et al., 1990*).

We anticipated that Hsp90 would be required at increased levels for robust experimental growth in diamide, nitrogen starvation, ethanol, and high temperature (*Gasch et al., 2000*) based on the concept that cells increase expression level of genes in conditions where those gene products are needed at higher concentration. Consistent with this concept, reduced Hsp90 levels cause a marked decrease in growth rate at 37°C. However, Hsp90 protein levels had smaller impacts on growth rates under the other stress conditions, indicating that reliance on overall Hsp90 function does not increase dramatically in these conditions.

We quantified the growth rates of all Hsp90 single-mutant variants in each of the stress conditions as selection coefficients (*Figure 2—source data 1*, *Figure 2—figure supplements 1–5*). We could clearly differentiate between the selection coefficients of wild-type synonyms and stop codons in all conditions (*Figure 2B*, *Figure 2—figure supplement 6*), and we normalized to these classes of mutations to facilitate comparisons between each condition (*Figure 2—figure supplement 7*). Of note, the observed selection coefficients of wild-type synonyms varied more in conditions of high temperature and diamide stress compared to standard (*Figure 2—figure supplement 8A,B*). We also note greater variation in the selection coefficients of barcodes for the same codon in the diamide and high temperature conditions (*Figure 2—figure supplement 8C*). We conclude that diamide and elevated temperature provided greater noise in our selection coefficient measurements. To take into account differences in signal to noise for each condition, we either averaged over large numbers of mutations or categorized selection coefficients as wild-type-like, strongly

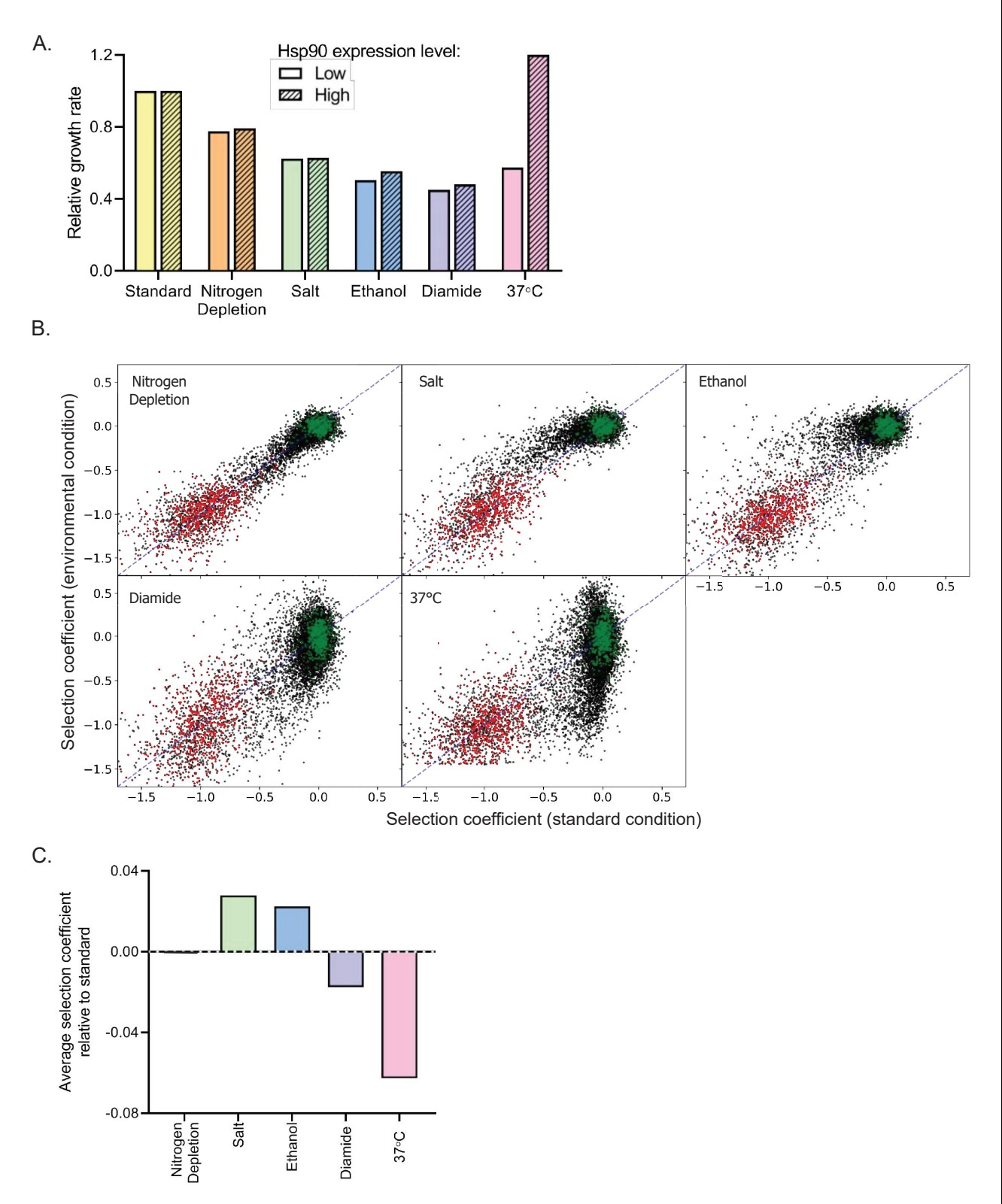

**Figure 2.** Impact of environmental stresses on yeast growth rates and selection on Hsp90 sequence. (A) Growth rate of yeast with normal and reduced expression of Hsp90 protein in standard and stress conditions based on individual growth curves. Growth rates are normalized to growth in standard conditions with reduced Hsp90 expression. (B) Selection coefficients of all Hsp90 amino acid variants in stress conditions compared to standard conditions (see *Figure 2—source data 1*). Wild-type synonyms are shown in green and stop codons are shown in red. Selection coefficients were

*Figure 2 continued on next page*

*Figure 2 continued*

scaled to null (s = −1) for the average stop codon and neutral (s = 0) for the average wild type. The diagonal is indicated by the blue dashed line. (**C**) The average selection coefficient of all mutations relative to standard conditions, a metric of the strength of selection acting on Hsp90 sequence, in each stress condition.

The online version of this article includes the following source data and figure supplement(s) for figure 2:

**Source data 1.** Sequencing counts and selection coefficients for each individual amino acid change across amino acids 2–709 of Hsp90 in Nitrogen Depletion, Salt, Ethanol, Diamide and 37°C.
**Figure supplement 1.** Heatmap representation of the fitness map observed for single amino acid changes of Hsp90 in nitrogen depletion.
**Figure supplement 2.** Heatmap representation of the fitness map observed for single amino acid changes of Hsp90 in salt.
**Figure supplement 3.** Heatmap representation of the fitness map observed for single amino acid changes of Hsp90 in ethanol.
**Figure supplement 4.** Heatmap representation of the fitness map observed for single amino acid changes of Hsp90 in diamide.
**Figure supplement 5.** Heatmap representation of the fitness map observed for single amino acid changes of Hsp90 at 37°C.
**Figure supplement 6.** Distribution of selection coefficients for non-synonymous mutations (black), wild-type synonyms (green), and stops (red) in each environmental condition.
**Figure supplement 7.** Distribution of selection coefficients in each environmental condition.
**Figure supplement 8.** Analysis of variation in wild-type synonym selection coefficients.
**Figure supplement 9.** Distribution of the difference between selection coefficients of each mutation in each stress condition and the same mutation in standard conditions.

deleterious, intermediate, or beneficial based on the distribution of wild-type synonyms and stop codons in each condition (see Materials and methods and *Figure 2—figure supplement 7*).

We compared selection coefficients of each Hsp90 variant in each stress condition to standard condition (*Figure 2B&C*). The stresses of 37°C and diamide tend to exaggerate the growth defects of many mutants compared to standard conditions, whereas high salt and ethanol tend to rescue growth defects (*Figure 2B&C* and *Figure 2—figure supplement 9*). According to the theory of metabolic flux (*Dykhuizen et al., 1987*; *Kacser and Burns, 1981*), gene products that are rate limiting for growth will be subject to the strongest selection. Accordingly, the relationship between Hsp90 function and growth rate should largely determine the strength of selection acting on Hsp90 sequence. Conditions where Hsp90 function is more directly linked to growth rate would be more sensitive to Hsp90 mutations than conditions where Hsp90 function can be reduced without changing growth rates (*Bershtein et al., 2013*; *Jiang et al., 2013*). The average selection coefficients are more deleterious in diamide and temperature stress compared to standard conditions. These findings are consistent with heat and diamide stresses causing a growth limiting increase in unfolded Hsp90 clients that is rate limiting for growth. In contrast, the average selection coefficients are less deleterious in ethanol and salt stress than in standard conditions, which suggests a decrease in the demand for Hsp90 function in these conditions. Due to the complex role Hsp90 plays in diverse signaling pathways in the cell, the different environmental stresses may differentially impact subsets of client proteins that cause distinct selection pressures on Hsp90 function.

## Structural analyses of environmental responsive positions

Altering environmental conditions had a pervasive influence on mutational effects along the sequence of Hsp90 (*Figure 3A* and *Figure 3—figure supplement 1*). We structurally mapped the average selection coefficient of each position in each condition relative to standard conditions as a measure of the sensitivity to mutation of each position under each environmental stress (*Figure 3A*, *Figure 3—source data 1*). Many positions had mutational profiles that were responsive to a range of environments. Environmentally responsive positions with large changes in average selection coefficient in at least three conditions are highlighted on the Hsp90 structure in green in *Figure 3B*. Unlike the critical positions that cluster around the ATP binding site (*Figure 1C*), the environmentally responsive positions are located throughout all domains of Hsp90. Similar to critical residues, environmentally responsive positions are more conserved in nature compared to other positions in Hsp90 (*Figure 3C*), suggesting that the suite of experimental stress conditions tested captured aspects of natural selection pressures on Hsp90 sequence.

Hsp90 positions with environmentally responsive selection coefficients were enriched in binding contacts with clients, co-chaperones and intramolecular Hsp90 contacts involved in transient conformational changes (*Figure 3D* and *Figure 3—figure supplement 2A*). About 65% of the

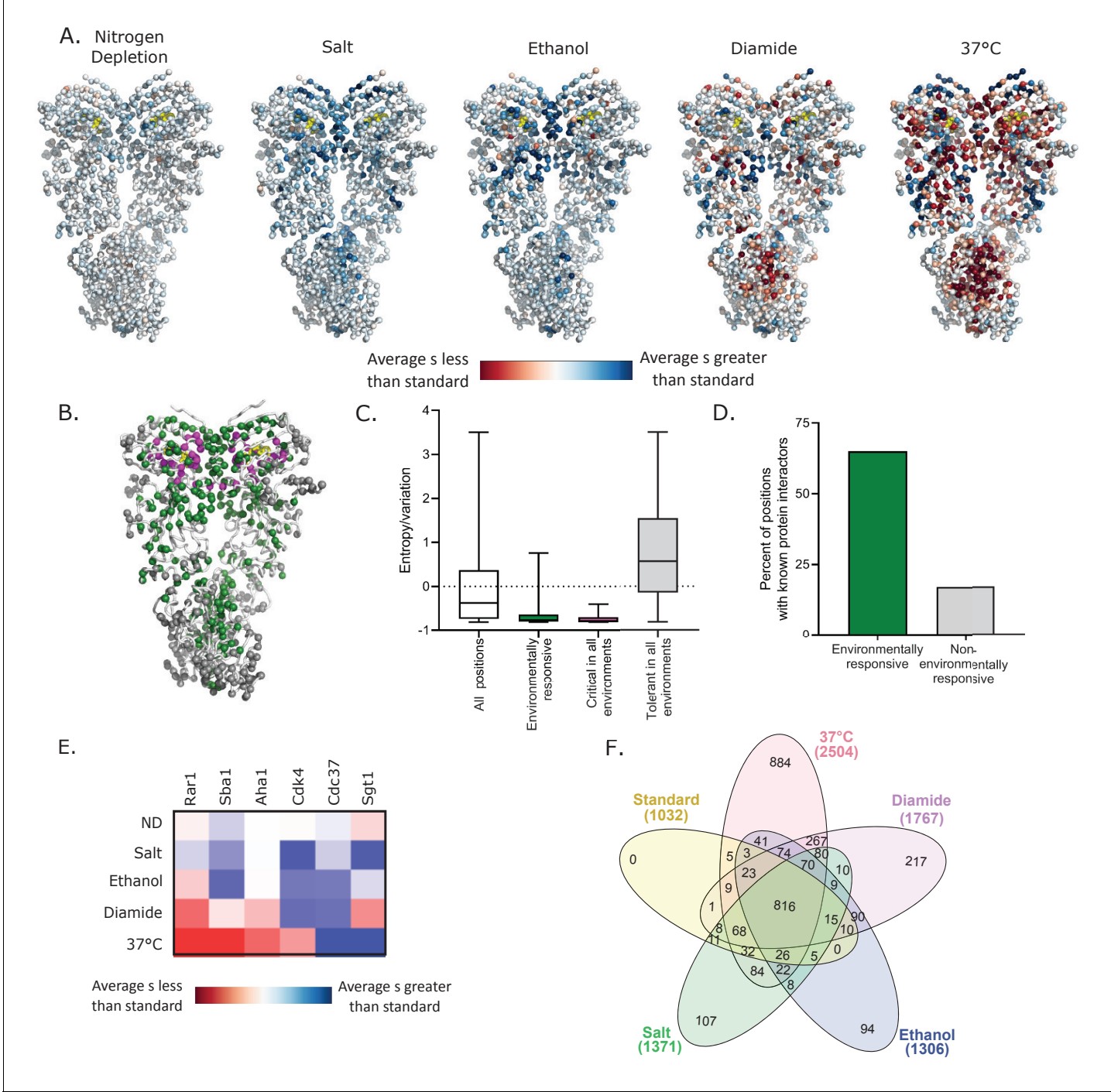

**Figure 3.** Environmental stresses place distinct selection pressures on Hsp90. (A) The average selection coefficient (s) at each position relative to standard conditions was mapped onto Hsp90 structure (*Ali et al., 2006*) (See *Figure 3—source data 1*). (B) Structural images indicating the location of positions that are critical for Hsp90 function in all conditions (magenta), positions that are environmentally responsive (ER) (green), and positions that are tolerant in all environments (gray). Critical residues have mean selection coefficients that are null-like in all environments. ER positions have mean selection coefficients that differed from standard in three or more environments by an amount greater than one standard deviation of wild-type synonyms. Tolerant residues are not shifted more than this cutoff in any environment. (C) For different classes of positions, evolutionary variation was calculated as amino acid entropy at each position in Hsp90 sequences from diverse eukaryotes. Distributions are significantly different as measured by a two-sample Kolmogorov-Smirnov (KS) (All positions vs. ER: N = 678, 137, p<0.0001, D = 0.39; All positions vs. critical: N = 678, 27, p<0.0001, D = 0.57; All positions vs. tolerant: N = 678, 136, p<0.0001, D = 0.38) (D) Fraction of different classes of mutations located at contact sites with binding partners. p<0.0001 (E) A heatmap of the average selection coefficient for all positions at the stated interfaces relative to standard conditions in each

*Figure 3 continued on next page*

*Figure 3 continued*

environment. (F) Venn diagram of deleterious mutations in different environmental conditions (*Heberle et al., 2015*). Total number of deleterious mutants in each condition are stated in parentheses.

The online version of this article includes the following source data and figure supplement(s) for figure 3:

**Source data 1.** Average selection coefficient (excluding stops) at each position of Hsp90 in each environmental condition relative to the average selection coefficient in standard conditions.

**Figure supplement 1.** Heatmap representation of the average selection coefficient (s) at each position in each environmental condition relative to the average selection coefficient at the same position in standard conditions.

**Figure supplement 2.** Environmentally responsive Hsp90 positions are enriched in binding contacts.

environmentally responsive residues have been identified either structurally or genetically as interacting with binding partners (*Ali et al., 2006*; *Bohen and Yamamoto, 1993*; *Genest et al., 2013*; *Hagn et al., 2011*; *Hawle et al., 2006*; *Kravats et al., 2018*; *Lorenz et al., 2014*; *Meyer et al., 2003*; *Meyer et al., 2004*; *Nathan and Lindquist, 1995*; *Retzlaff et al., 2009*; *Roe et al., 2004*; *Verba et al., 2016*; *Zhang et al., 2010*), compared to about 15% of positions that were not responsive to stress conditions. This analysis was performed on the small subset of clients and cochaperones with known Hsp90-binding sites. While ATP binding and hydrolysis are the main structural determinants that constrain fitness in standard growth conditions, client and co-chaperone interactions have a larger impact on experimental fitness under stress conditions. Although the mean selection coefficients of mutations at the known client and co-chaperone binding sites are responsive to changes in environment, the direction of the shift of growth rate compared to standard conditions depends on the specific binding partner and environment (*Figure 3E* and *Figure 3—figure supplement 2B*). This suggests that different environments place unique functional demands on Hsp90 that may be mediated by the relative affinities of different clients and co-chaperones. Consistent with these observations, we hypothesize that Hsp90 client priority is determined by relative binding affinity and that Hsp90 mutations can reprioritize clients that in turn impacts many signaling pathways.

## Constraint of mutational sensitivity at high temperature

We find that different environmental conditions lead to distinct selection on Hsp90 based on the number of beneficial and deleterious variants in each condition, including elevated temperature placing the greatest purifying selection pressure on Hsp90. Of the 2504 variants of Hsp90 that are deleterious when grown at 37°C, 884 of them (~35%) are deleterious only in this condition (*Figure 3F*). We defined mutants that confer temperature sensitive (*ts*) growth phenotypes on cells as variants with selection coefficients within the distribution of wild-type synonyms in standard conditions and that of stop codons at 37°C. Based on this definition, 663 Hsp90 amino acid changes (roughly 5% of possible changes) were found to be temperature sensitive (*Figure 4A*, *Figure 4—source data 1*). We sought to understand the physical underpinnings of this large set of Hsp90 *ts* mutations.

We examined Hsp90 *ts* mutations for structural and physical patterns. We found that *ts* mutations tended to concentrate at certain amino acid positions of Hsp90 (*Figure 4B*). The clustering of *ts* mutations was significant compared to random simulations. Positions with greater than four *ts* mutations were spread across all three domains of Hsp90 (*Figure 4C*) with the largest cluster occurring in the C domain of Hsp90. The C domain forms a constitutive homodimer that is critical for function (*Wayne and Bolon, 2007*). Of note, homo-oligomerization domains may have a larger *ts* potential because all subunits contribute to folding and dimerization essentially multiplying the impacts of mutations (*Lynch, 2013*). To explore the physical underpinnings of *ts* mutations, we examined if they were buried in the structure or surface exposed. Mutations at buried residues tend to have a larger impact on protein folding energy compared to surface residues (*Chakravarty and Varadarajan, 1999*). Consistent with the idea that many *ts* mutations may disrupt protein folding at elevated temperature, substitutions that confer a *ts* phenotype are enriched in buried residues (*Figure 4D*). Also consistent with this idea, *ts* mutations tend to have negative Blosum scores (*Henikoff and Henikoff, 1992*; *Figure 4E*), a hallmark of disruptive amino acid changes.

Because growth at elevated temperatures requires higher levels of Hsp90 protein (*Borkovich et al., 1989*), some *ts* mutations are likely due to a reduced function that is enough for

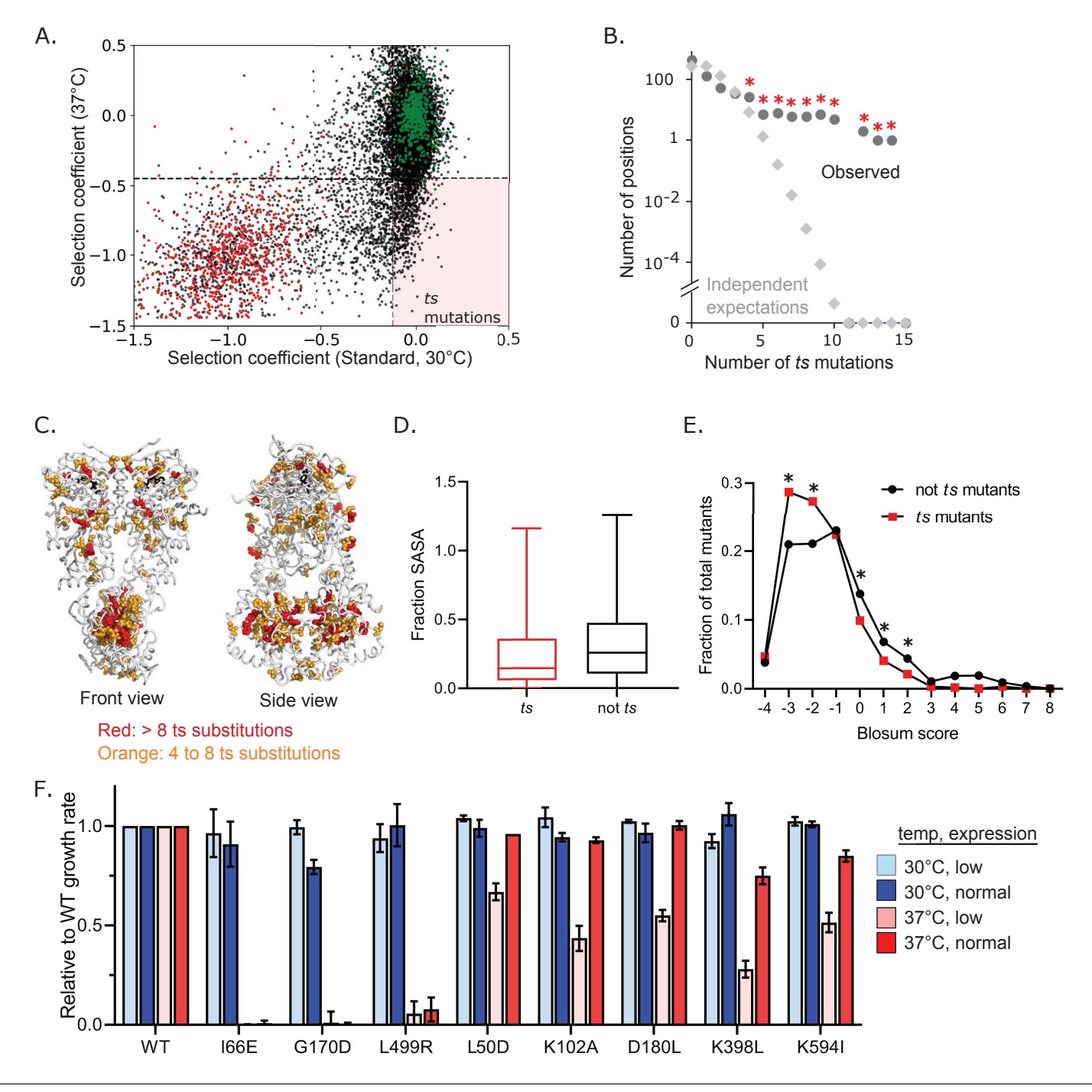

**Figure 4.** Abundance and mechanism of temperature-sensitive (*ts*) mutations in Hsp90. (**A**) *Ts* variants were identified that supported WT-like growth at 30°C, but were null-like at 37°C in bulk competitions. WT synonyms are shown in green and stops in red. The horizonal dashed line corresponds to y = −0.47, the upper limit of stops at 37°C and the vertical dashed line corresponds to x = −0.11, the lower limit of WT synonyms in standard conditions. The pink-shaded quadrant highlights *ts* mutations. All *ts* mutants are listed in *Figure 4—source data 1*. (**B**) Distribution of the number of observed *ts* mutations at the same positions of Hsp90 (•) is much greater than expected if they had occurred independently (◇). Independent expectations were calculated as the probability of the stated number of mutations occurring at the same position by chance. * Indicates observations that were significantly (p<0.01) greater than independent expectations based on random simulations and one-tailed t-tests. (**C**) Mapping positions with multiple *ts* variants onto Hsp90 structure. ATP is shown in black. (**D**) Solvent accessible surface area (SASA) of *ts* mutants compared to non-*ts* mutants. A two sample KS test showed significant differences in distributions (N = 663, 11762, p<0.0001, D = 0.1735) (**E**) Amino acid similarity to the wild type was estimated as the Blosum score (*Henikoff and Henikoff, 1992*) for *ts* and non-*ts* variants. A two-proportion z-test was performed on each pair for each

*Figure 4 continued on next page*

Figure 4 continued

Blosum score and their p-values were adjusted using Benjamini-Hochberg adaptive step-up procedure. * Indicates p<0.05 F. Growth rate of a panel of individual Hsp90 *ts* variants analyzed in isolation.

The online version of this article includes the following source data for figure 4:

**Source data 1.** List of all temperature-sensitive mutants and associated selection coefficients.

growth at standard temperature, but is insufficient at 37°C (*Nathan and Lindquist, 1995*). We reasoned that we could distinguish these mutants by examining how growth rate depended on the expression levels of Hsp90. We expect that destabilizing mutants that cause Hsp90 to unfold at elevated temperature would not support efficient growth at 37°C independent of expression levels. In contrast, we expect mutants that reduce Hsp90 function to exhibit an expression-dependent growth defect at 37°C. We tested a panel of *ts* mutations identified in the bulk competitions at high and low expression levels (*Figure 4F*). The dependence of growth rate at 37°C on expression level varied for different Hsp90 *ts* variants. The I66E, G170D and L499R Hsp90 mutants have no activity at 37°C irrespective of expression levels. These disruptive substitutions at buried positions likely destabilize the structure of Hsp90. In contrast, increasing the Hsp90 expression levels at least partially rescued the growth defect for five *ts* variants (L50D, K102A, D180L, K398L, K594I), indicating that these variants do not provide enough Hsp90 function for robust growth at elevated temperature. All five of these expression-dependent *ts* variants were located at surface positions. Thus, for the *ts* mutants we tested individually, we see a correlation between location of the mutation and type of *ts* mutation. Destabilizing mutations tend to be buried and mutants with reduced function tend to be surface exposed, indicating that the location of *ts* mutations can delineate these different mechanistic classes.

## Hsp90 potential for adaptation to environmental stress

Numerous Hsp90 variants provided a growth benefit compared to the wild-type sequence in stress conditions. The largest number of beneficial variants in Hsp90 occurred in high temperature and diamide conditions (*Figure 5A*, *Figure 5—source data 1*). Multiple lines of evidence indicate that these mutants are truly beneficial variants and not simply measurement noise. First, the beneficial amino acids generally exhibited consistent selection coefficients among synonymous variants (*Figure 5—figure supplement 1A*). Second, beneficial mutants in diamide and high temperature tend to cluster at certain positions (*Figure 5B*), which would not be expected for noise. Finally, we confirmed the increased growth rate at elevated temperature of a panel of variants analyzed in isolation (*Figure 5—figure supplement 1B*). The fact that beneficial mutations in elevated temperature and diamide often clustered at specific positions in Hsp90 indicates that the wild-type amino acids at these positions are far from optimum for growth in these conditions. In contrast, the apparent beneficial mutations in other conditions did not tend to cluster at specific positions (*Figure 5—figure supplement 2*).

To obtain a more general picture of the potential for adaptation derived from the full fitness distributions, we used Fisher's Geometric model (FGM) (*Fisher, 1930*). According to FGM, populations evolve in an *n*-dimensional phenotypic space, through random single-step mutations, and any such mutation that brings the population closer to the optimum is considered beneficial. An intuitive hypothesis derived from FGM is that the potential for adaptation in a given environment (i.e. is the availability of beneficial mutations) depends on the distance to the optimum. In order to estimate the distance to the optimum *d*, we adopted the approach by Martin and Lenormand and fitted a displaced gamma distribution to the neutral and beneficial mutations for each environment (*Martin and Lenormand, 2006*). We observed that the yeast populations were furthest from the optimum in elevated temperature and diamide (*d* = 0.072 and 0.05, respectively), followed by nitrogen deprivation (*d* = 0.023), high salinity and ethanol (*d* = 0.021) and standard (*d* = 0.014). This suggests that exposure to elevated temperature and diamide results in the largest potential for adaptation and is consistent with the observation of the largest proportions of beneficial mutations in these environments. Interestingly, previous results from a 9-amino-acid region in Hsp90 indicated that there was very little potential for adaptation at high temperature (36°C) as compared with high salinity (*Hietpas et al., 2013*). This apparent contradiction between results from the full Hsp90

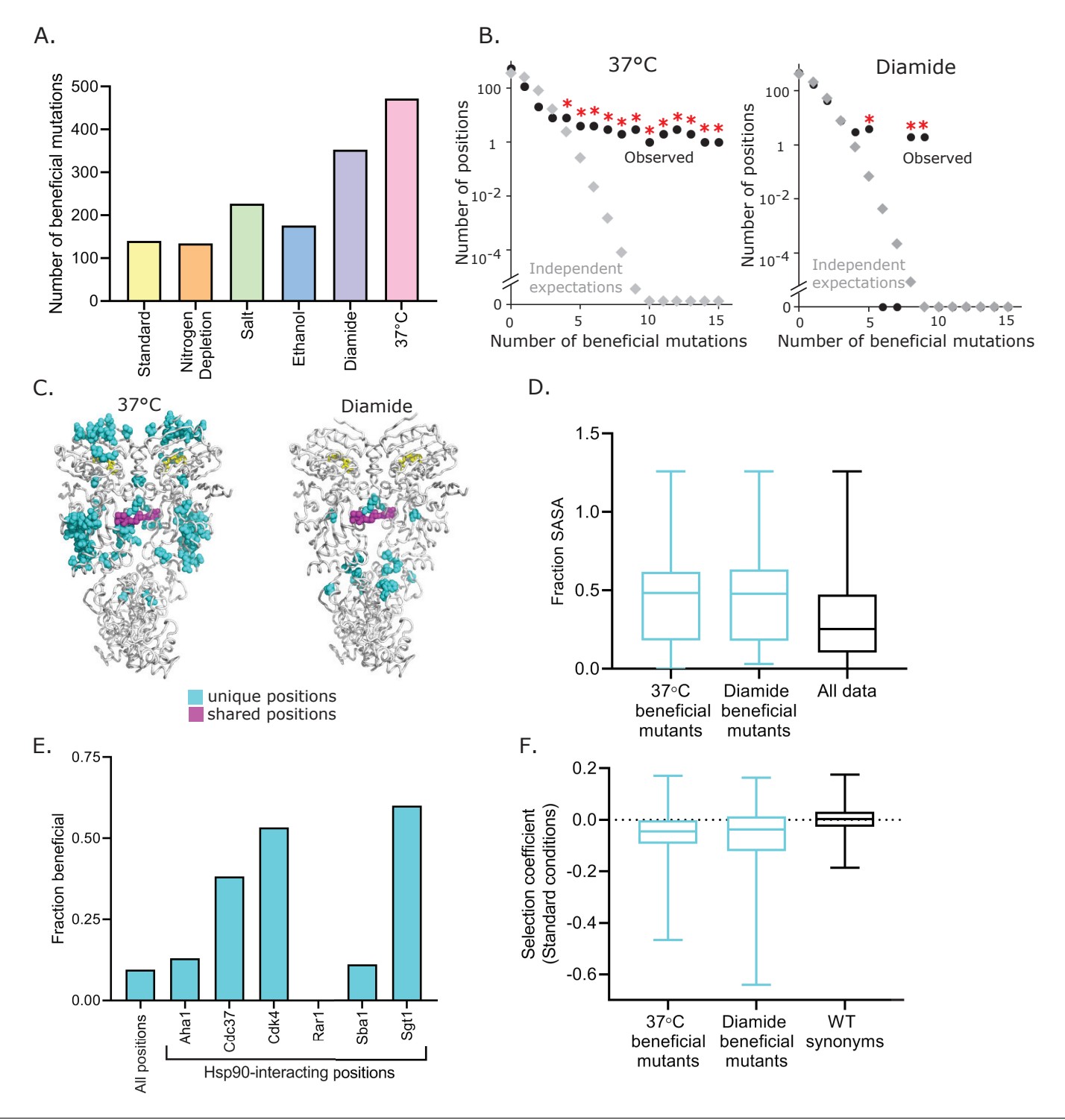

**Figure 5.** Beneficial variants in diamide and elevated temperature conditions. (A) Number of beneficial mutations identified in each condition based on selection coefficients more than two standard deviations greater than wild-type synonyms. Beneficial mutants at 37°C and in diamide are listed in *Figure 5—source data 1*. (B) Distribution of the number of beneficial mutations at the same position in both 37°C (left) and diamide (right) conditions (•) is greater than expected if they had occurred independently (◊). Independent expectations were calculated as the probability of the stated number of mutations occurring at the same position by chance. * Indicates observations that were significantly (p<0.01) greater than independent expectations based on random simulations and one-tailed t-tests. (C) Location of positions with four or more beneficial mutations. Positions that are unique to diamide or 37°C are shown in cyan and two shared positions are shown in magenta. (D) The solvent accessible surface area (SASA) of beneficial

*Figure 5 continued on next page*

*Figure 5 continued*

mutations at 37°C and in diamide compared to all mutations. Distributions are significantly different as measured by a two-sample KS test (37°C vs. all data, N = 270, 12393, p<0.0001, D = 0.2851; Diamide vs. all data, N = 60,12393, p<0.0001, D = 0.3465) (E) The fraction of Hsp90 positions at interfaces that were beneficial in 37°C and diamide conditions. (F) Selection coefficients in standard conditions for beneficial mutations at 37°C and in diamide compared to wild-type synonyms. (KS test; 37°C vs. WT synonyms: N = 463, 660, p<0.0001, D = 0.3281; Diamide vs. WT synonyms: N = 353, 660, p<0.0001, D = 0.3809).

The online version of this article includes the following source data and figure supplement(s) for figure 5:

**Source data 1.** List of all beneficial mutants and associated selection coefficients at 37°C and in diamide.
**Figure supplement 1.** Validation of beneficial mutants at37°C.
**Figure supplement 2.** Distribution of the number of beneficial mutations at the same position in standard, nitrogen depletion, salt, and ethanol conditions.
**Figure supplement 3.** Selection coefficients in standard conditions for all wild-type-like mutations at 37°C and in diamide.
**Figure supplement 4.** Synonymous mutations at the beginning of Hsp90 have strong beneficial growth effects.

sequence and the 582–590 region indicates that a specific region of the protein may be already close to its functional optimum in a specific environment, whereas there is ample opportunity for adaptation when the whole protein sequence is considered.

In diamide and elevated temperature, the clustered beneficial positions were almost entirely located in the ATP-binding domain and the middle domain (*Figure 5C*), both of which make extensive contacts with clients and co-chaperones (*Ali et al., 2006*; *Meyer et al., 2003*; *Meyer et al., 2004*; *Roe et al., 2004*; *Verba et al., 2016*; *Zhang et al., 2010*). Beneficial mutations in elevated temperature and diamide conditions were preferentially located on the surface of Hsp90 (*Figure 5D*) at positions accessible to binding partners. Analyses of available Hsp90 complexes indicate that beneficial positions were disproportionately located at known interfaces with co-chaperones and clients (*Figure 5E*). Clustered beneficial mutations are consistent with disruptive mechanisms because a number of different amino acid changes can lead to disruptions, whereas a gain of function is usually mediated by specific amino acid changes. Amino acids that are beneficial in diamide and elevated temperature tend to exhibit deleterious effects in standard conditions (*Figure 5F*), consistent with a cost of adaptation. In comparison, wild-type-like mutations in diamide and high temperature tend to exhibit wild-type-like fitness in standard conditions (*Figure 5—figure supplement 3*). We conjecture that the clustered beneficial mutations are at positions that mediate the binding affinity of subsets of clients and co-chaperones and that disruptive mutations at these positions can lead to re-prioritization of multiple clients. The priority or efficiency of Hsp90 for sets of clients can in turn impact most aspects of physiology because Hsp90 clients include hundreds of kinases that influence virtually every aspect of cell biology.

In the first seven amino acids of Hsp90, we noted both a large variation in the selection coefficients of synonymous mutations at elevated temperature and that many nonsynonymous substitutions at these positions generated strong beneficial effects (*Figure 5—figure supplement 4A,B*). Synonymous mutations at these positions were only strongly beneficial at high temperature where Hsp90 protein levels are limiting for growth. Analysis of an individual clone confirms that a synonymous mutation at the beginning of Hsp90 that was beneficial at high temperature was expressed at higher level in our plasmid system (*Figure 5—figure supplement 4C,D*). These results are consistent with a large body of research showing that mRNA structure near the beginning of coding regions often impacts translation efficiency (*Li, 2015*; *Plotkin and Kudla, 2011*; *Tuller et al., 2010*), and that adaptations can be mediated by changes in expression levels (*Lang and Desai, 2014*).

## Natural selection favors Hsp90 variants that are robust to environment

We next examined how experimental protein fitness maps compared with the diversity of Hsp90 sequences in current eukaryotes. We analyzed Hsp90 diversity in a set of 267 sequences from organisms that broadly span across eukaryotes. We identified 1750 amino acid differences in total that were located at 499 positions in Hsp90. We examined the experimental growth effects of the subset of amino acids that were observed in nature. While the overall distribution of selection coefficients in all conditions was bimodal with peaks around neutral (s = 0) and null (s = −1), the natural amino acids were unimodal with a peak centered near neutral (*Figure 6A*, *Figure 6—source data 1*). The vast majority of natural amino acids had wild-type-like fitness in all conditions studied here

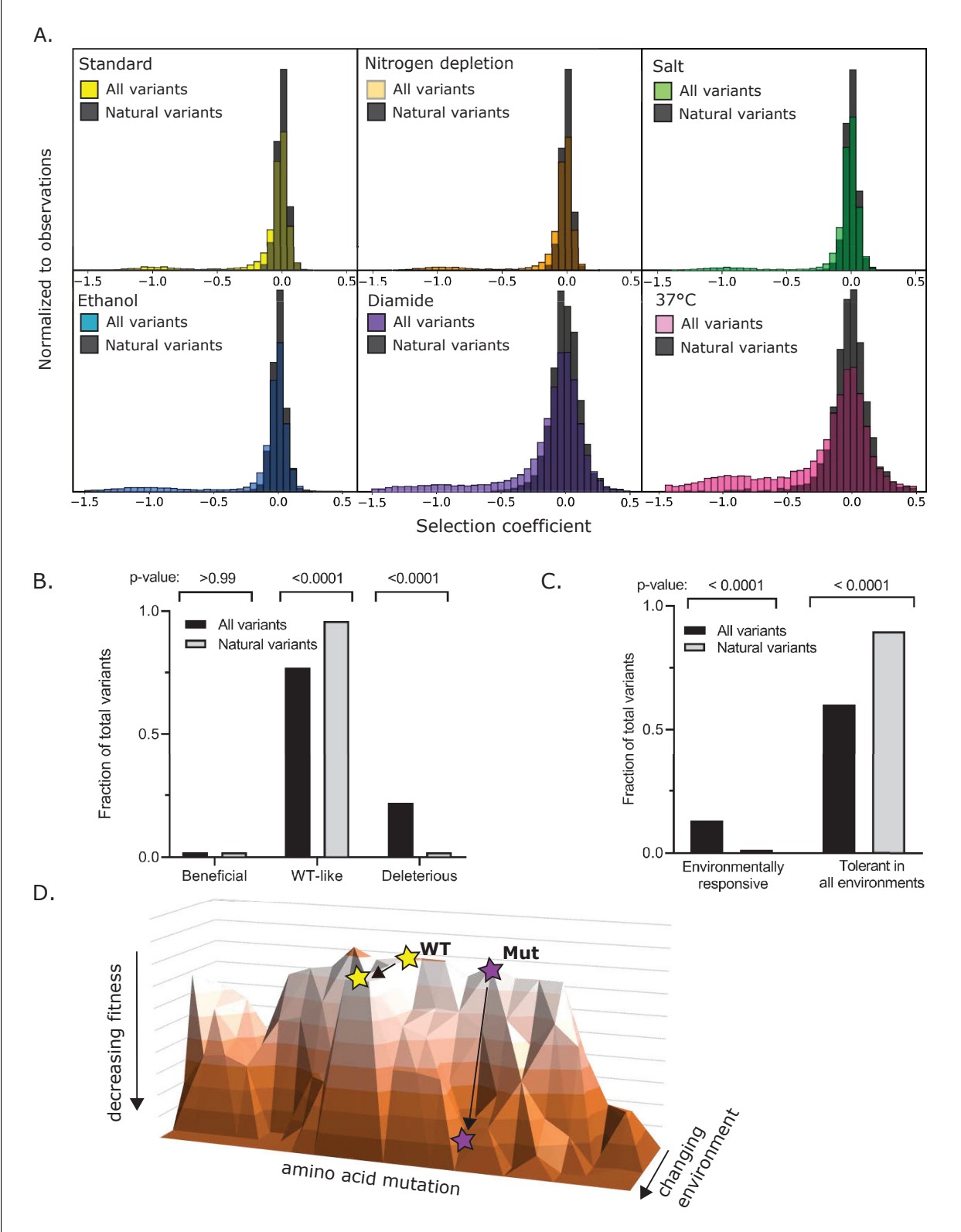

**Figure 6.** Experimental growth effects of natural amino acid variants of Hsp90. (**A**) The distribution of selection coefficients of natural variants compared to all variants in each environmental condition (see *Figure 6—source data 1*). (**B**) Across all environments, the fraction of natural variants compared to all variants that were beneficial, wild-type-like, or deleterious. (**C**) The fraction of natural variants compared to all variants that were environmentally

*Figure 6 continued on next page*

*Figure 6 continued*

responsive or tolerant in all environments. Categories were defined as in *Figure 3B*. (D) Landscape model indicating that natural variants of Hsp90 tend to support robust growth under a variety of stress conditions.

The online version of this article includes the following source data and figure supplement(s) for figure 6:

**Source data 1.** List of selection coefficients of all natural variants of Hsp90 in all environmental conditions.

**Figure supplement 1.** Hsp90 expression in the Hsp90 shutoff yeast strain harboring either wild-type Hsp90 under the constitutive ADH promoter or a null plasmid (no insert).

**Figure supplement 2.** Heatmap of Spearman's rank correlation coefficients (r) between molecular features (rows) and selection coefficients per mutation or per position (average selection coefficient for all amino acids at the position) for each environment (columns).

**Figure supplement 2—source data 1.** The Spearman's rank correlation coefficients (r) and associated p-values between molecular features and selection coefficients for each environment.

**Figure supplement 3.** Validation of yeast Hsp90-shutoff strain.

---

(*Figure 6B and C*). Whereas naturally occurring amino acids in Hsp90 were rarely deleterious in any experimental condition, they were similarly likely to provide a growth benefit compared to all possible amino acids (5%). This observation indicates that condition-dependent fitness benefits are not a major determinant of natural variation in Hsp90 sequences. Instead, our results indicate that natural selection has favored Hsp90 substitutions that are robust to multiple stressful conditions (*Figure 6D*). Beneficial mutations in heat and diamide indicate that there is room for improvement in Hsp90 function in individual conditions. The clustering of beneficial mutations at known binding interfaces suggests that the optimal binding affinity for partner proteins may depend on growth conditions. We propose that natural variants of Hsp90 have been selected for binding properties that are robust to different stresses rather than specific to individual conditions.

Epistasis may provide a compelling explanation for the naturally occurring amino acids that we observed with deleterious selection coefficients. Analyses of Hsp90 mutations in the context of likely ancestral states has demonstrated a few instances of historical substitutions with fitness effects that depend strongly on the Hsp90 sequence background (*Starr et al., 2018*). Indeed, many of the natural amino acids previously identified with strong epistasis (E7A, V23F, T13N) are in the small set of natural amino acids with deleterious effects in at least one condition. Further analyses of natural variants under diverse environmental conditions will likely provide insights into historical epistasis and will be the focus of future research.

## Discussion

In this study, we analyzed the protein-wide distribution of fitness effects of Hsp90 across standard and five stress conditions. We found that environment has a profound effect on the fates of Hsp90 mutations. Each environmental stress varies in the strength of selection on Hsp90 mutations; heat and diamide increase the strength of selection and ethanol and salt decrease the strength of selection. While proximity to ATP is the dominant functional constraint in standard conditions, the influence of client and co-chaperone interactions on growth rate dramatically increases under stress conditions. Additionally, beneficial mutations cluster at positions that mediate binding to clients and cochaperones. The fact that different Hsp90 binding partners have distinct environmental dependencies suggests that Hsp90 can reprioritize clients that in turn impacts many downstream signaling pathways.

Our results demonstrate that mutations to Hsp90 can have environment-dependent effects that are similar to the stress-induced changes to the function of wild-type Hsp90 that have been shown to contribute to new phenotypes (*Jarosz et al., 2010*). The low frequency of environment-dependent amino acids in Hsp90 from extant eukaryotes indicates that this type of evolutionary mechanism is rare relative to drift and other mechanisms shaping Hsp90 sequence diversity.

We observed distinct structural trends for mutations that provide environment-dependent costs and benefits. Many mutations in Hsp90 caused growth defects at elevated temperature where Hsp90 function is limiting for growth. These temperature-sensitive mutations tended to be buried and in the homodimerization domain, consistent with an increased requirement for folding stability at elevated temperatures. In contrast, beneficial mutations tended to be on the surface of Hsp90 and at contact sites with binding partners, suggesting that change-of-function mutations may be

predominantly governed by alterations to binding interactions. Mutations that disrupt binding to certain clients can lead to the re-prioritization of others, which, due to Hsp90's central role in numerous cellular pathways, has the potential to modify integral networks in response to stress. Once more comprehensive data is available on Hsp90-client binding sites, it may be possible to simulate this rewiring of cellular networks, providing insight into the causes of the beneficial mutations. However, presently, the large number of clients with unknown binding sites makes these analyses challenging. In the future, comparing Hsp90 client-interactomes (*Taipale et al., 2012*) may help delineate adaptive biochemical mechanisms.

Limitations:

Our experimental setup has limitations that we have tried to account for in our analyses and conclusions. For example, we measured the fitness effects of Hsp90 under artificially low expression where yeast growth rates are tightly coupled to function of Hsp90 in order to provide a sensitive readout of fitness defects (*Jiang et al., 2013*). Expression of Hsp90 under this promoter remains stable in the stresses tested (*Figure 6—figure supplement 1*). However, this defined promoter does not capture the native transcriptional regulation and may not fully recapitulate translational and post-translational regulation controlling *hsp82*. While these levels of regulation of Hsp90 are clearly important physiologically, the sensitive readouts of fitness that we measured appear to capture critical features of Hsp90 with regard to biochemical function and evolutionary mechanism. For example, virtually all deleterious mutants measured in this study under stress conditions appear to have also been subject to purifying selection in nature.

In addition, the experimental strain used in this study is deleted for the constitutively expressed paralog of *hsp82*, *hsc82. Hsp82* and *hsc82* are functionally overlapping, essential genes with 97% sequence identify (16 amino acid differences) that can compensate for each other's loss-of-function in normal growth conditions (*Girstmair et al., 2019*). The high sequence identity between the two paralogs indicates that they are both under similar selection pressure. Despite the high sequence identity, a number of distinct differences have been noted in stability, conformational cycles, and client interactomes. Experimental evidence indicates that *hsp82* is more stress-specific, and more stable to unfolding (*Girstmair et al., 2019*). Further efforts will be required to resolve how distinctions between Hsp90 paralogs contribute to function and selection.

Relationship to prior work:

A handful of studies have assessed the impact of environment on the fitness landscape of genes (*Dandage et al., 2018*; *Hietpas et al., 2013*; *Li and Zhang, 2018*; *Mavor et al., 2016*). For example, Dandage, Chakraborty and colleges investigated the effects of temperature and chemical chaperones on the fitness landscape of the Gentamicin resistance gene and found that protein stability and distance to the ligand binding site are the molecular properties with the strongest correlations with fitness (*Dandage et al., 2018*). To understand the strength of the molecular constraints on Hsp90 on a whole protein level, we performed similar analyses (*Figure 6—figure supplement 2*). Consistent with the Gentamicin study, we find the features that best correlate with fitness are protein stability and distance from the active site. The constraint of protein stability is the highest at 37°C, indicating increased dependence on stability at the higher temperature. In addition, distance from the ATP-binding site imposes strong molecular constraints on Hsp90, signifying the importance of ATP hydrolysis on Hsp90 function. While individual features correlate with fitness effects and show environmental dependence, single features are unable to capture the majority of observed variance in fitness effects, consistent with a complex set of physical properties that underlie fitness effects in both proteins.

In another study of the effect of environment on mutational fitness, Li and Zhang detected pervasive genotype-by-environment interactions between a yeast tRNA gene and environment (*Li and Zhang, 2018*). They found that the correlation of the fitness between mutations in each tested environment was linear such that the fitness landscape in one environment together with a change in slope could be used to accurately predict fitness effects in the second environment. In this study, we observed a large impact of environment on Hsp90 fitness; however, we observe many fitness effects that deviate from a linear relationship between environments. While linear models can predict the fitness of some mutations in different environments, it would not predict many of the types of mutations that are focuses of this study, such as mutations that exhibit an adaptive trade-off, those with beneficial effects in one environment that become deleterious in another. In addition, the linear model would not predict the large group of *ts* mutations with wild-type fitness in standard

conditions and null fitness at 37°C. As environmental-dependent protein fitness landscapes are analyzed for an increasing set of genes, it will provide opportunities to explore how different protein properties such as the number of binding partners may contribute to global trends.

Importantly, our results demonstrate that while mutations to Hsp90 can provide a growth advantage in specific environmental conditions, naturally occurring amino acids in Hsp90 tend to support robust growth over multiple stress conditions. The finding of beneficial mutations in Hsp90 in specific conditions suggests that similar long-term stresses in nature can lead to positive selection on Hsp90. Consistent with previous work (*Hietpas et al., 2013*), we found that experimentally beneficial mutations tended to have a fitness cost in alternate conditions (*Figure 5F*). This indicates that natural environments which fluctuate among different stresses would reduce or eliminate positive selection on Hsp90. Therefore, our results suggest that natural selection on Hsp90 sequence has predominantly been governed by strong purifying selection integrated over multiple stressful conditions. Taken together, these results support the hypothesis that natural populations might experience a so-called 'micro-evolutionary fitness seascape' (*Mustonen and Lässig, 2009*), in which rapidly fluctuating environments result in a distribution of quasi-neutral substitutions over evolutionary time scales.

# Materials and methods

## Key resources table

| Reagent type (species) or resource | Designation | Source or reference | Identifiers | Additional information |
|---|---|---|---|---|
| Gene (*Saccharomyces cerevisiae*) | *hsp82* | *Saccharomyces* Genome Database | SGD: S000006161 | Hsp90 chaperone |
| Antibody | anti-Hsp90 α/β (Mouse monoclonal) | Cayman chemical; Cat# 10011439 | RRID:AB_10349777 | WB (1:3000) |
| Recombinant DNA reagent | Barcoded Hsp90 plasmid library | This paper | | See Materials and methods for library construction |
| Recombinant DNA reagent | p414ADH Δter plasmid | PMID: 23825969 | | |
| Recombinant DNA reagent | p414GPD plasmid | PMID: 23825969 | | |
| Commercial assay or kit | BCA Protein Assay Kit | Pierce | Cat #23227 | |
| Commercial assay or kit | KAPA SYBR FAST qPCR Master Mix | Kapa Biosystems | KK4600 | |
| Chemical compound, drug | Diamide | Sigma Aldrich | D3648 | |
| Software, algorithm | Barcode – Hsp90 ORF assembly | This paper | https://github.com/JuliaFlynn/Barcode_ORF_assembly | Associates barcodes with open reading frame mutations from paired end sequencing data (*Flynn, 2020a*; copy archived at https://github.com/elifesciences-publications/Barcode_ORF_assembly) |
| Software, algorithm | Tabulate Hsp90 counts | This paper | https://github.com/JuliaFlynn/Tabulate_counts | Counts Hsp90 alleles from raw fastq files and barcode_orf assembly file (*Flynn, 2020b*; copy archived at https://github.com/JuliaFlynn/Tabulate_counts) |
| Software, algorithm | EmpiricIST | PMID: 30127529 | https://github.com/Matu2083/empiricIST | Estimates selection coefficients based on the MCMC approach |
| Sequenced-based reagent | Sequencing primers | This paper | | See *Supplementary file 1* |

*Continued on next page*

*Continued*

| Reagent type (species) or resource | Designation | Source or reference | Identifiers | Additional information |
|---|---|---|---|---|
| Sequenced-based reagent | Site-directed mutagenesis primers | This paper | | See *Supplementary file 1* |
| Sequenced-based reagent | Library construction oligomers | This paper | | Available upon request |

### Generating mutant libraries

A library of Hsp90 genes was saturated with single point mutations using oligos containing NNN codons as previously described (*Hietpas et al., 2012*). The resulting library was pooled into 12 separate 60 amino acid long sub-libraries (amino acids 1–60, 61–120 etc.) and combined via Gibson Assembly (NEB) with a linearized p414ADHΔter Hsp90 destination vector, a low copy number plasmid with the *trp1* selectable marker. To simplify sequencing steps during bulk competition, each variant of the library was tagged with a unique barcode. For each 60 amino acid sub-library, a pool of DNA constructs containing a randomized 18 bp barcode sequence (N18) was cloned 200 nt downstream from the Hsp90 stop codon via restriction digestion, ligation, and transformation into chemically competent *E. coli* with the goal of each mutant being represented by 10–20 unique barcodes.

### Barcode association of library variants

We added barcodes and associated them with Hsp90 variants essentially as previously described (*Starr et al., 2018*). To associate barcodes with Hsp90 variants, we performed paired-end sequencing of each 60 amino acid sub-library using a primer that reads the N18 barcode in one read and a primer unique to each sub-library that anneals upstream of the region containing mutations. To facilitate efficient Illumina sequencing, we generated PCR products that were less than 1 kb in length for sequencing. We created shorter PCR products by generating plasmids with regions removed between the randomized regions and the barcode. To remove regions from the plasmids, we performed restriction digest with two unique enzymes, followed by blunt ending with T4 DNA polymerase (NEB) and plasmid ligation at a low concentration (3 ng/µL) to favor circularization over bimolecular ligations. The resulting DNA was re-linearized by restriction digest, and amplified with 11 cycles of PCR to generate products for Illumina sequencing. The resulting PCR products were sequenced using an Illumina MiSeq instrument with asymmetric reads of 50 bases for Read1 (barcode) and 250 bases for Read2 (Hsp90 sequence). After filtering low-quality reads (Phred scores < 10), the data was organized by barcode sequence. For each barcode that was read more than three times, we generated a consensus of the Hsp90 sequence that we compared to wild type to call mutations. Of note, the building of consensus of at least three independent reads reduces the chance that errors will lead to mistaken variant identity because the same misread would have to occur in the majority of these reads.

### Bulk growth competitions

Equal molar quantities of each sub-library were mixed to form a pool of DNA containing the entire Hsp90 library with each codon variant present at similar concentration. The plasmid library was transformed using the lithium acetate procedure into the DBY288 Hsp90 shutoff strain of *S. cerevisiae* which has both genomic paralogs of Hsp90 (*hsp82* and *hsc82*) deleted and a chromosomal copy of *hsp82* under a galactose-dependent promoter inserted (*can1-100 ade2-1 his3-11,15 leu2-3, 12 trp1-1, ura3-1 hsp82::leu2 hsc82::leu2 ho::pgals-hsp82-his3*) essentially as previously described (*Jiang et al., 2013*). Sufficient transformation reactions were performed to attain ~5 million independent yeast transformants representing a fivefold sampling for the average barcode and 50 to 100-fold sampling for the average codon variant. Following 12 hr of recovery in SRGal (synthetic 1% raffinose and 1% galactose) media, transformed cells were washed five times in SRGal-W media (SRGal lacking tryptophan to select for the presence of the Hsp90 variant plasmid) to remove extracellular DNA, and grown in SRGal-W media at 30˚C for 48 hr with repeated dilution to maintain the cells in log phase of growth. This yeast library was supplemented with 20% glycerol, aliquoted and slowly frozen in a −80˚C freezer.

For each competition experiment, an aliquot of the frozen yeast library cells was thawed at 37°C. Viability of the cells was accessed before and after freezing and was determined to be greater than 90% with this slow freeze, quick thaw procedure. Thawed cells were amplified in SRGal-W for 24 hr, and then shifted to shutoff conditions by centrifugation, washing, and resuspension in 300 mL of synthetic dextrose lacking tryptophan (SD-W) for 12 hr at 30°C. At this time, cells containing a null-rescue plasmid had stopped growing and Hsp90 was undetectable by western blot (*Figure 6—figure supplement 3*). At this point, cells were split and transferred to different conditions including: Standard (SD-W, 30°C), Nitrogen depletion (SD-W with limiting amounts of ammonium sulfate, 0.0125%, 30°C), Salt (SD-W with 0.8 M NaCl, 30°C), Ethanol (SD-W with 7.5% ethanol, 30°C), Diamide (SD-W with 0.85 mM diamide, 30°C), or high temperature (SD-W, 37°C). We collected samples of ~$10^8$ cells at eight time points over a period of 36 hr and stored them at −80°C. Cultures were maintained in log phase by regular dilution with fresh media every 6–10 hr to maintain a population size of $10^8$–$10^9$ cells in order to prevent population bottlenecks relative to sample diversity. Bulk competition from the standard condition were conducted in technical duplicates from the frozen yeast library.

## DNA preparation and sequencing

We isolated plasmid DNA from each bulk competition time point as described (*Jiang et al., 2013*). Purified plasmid was linearized with AscI. Barcodes were amplified by 19 cycles of PCR using Phusion polymerase (NEB) and primers that add Illumina adapter sequences and an 8 bp identifier sequence used to distinguish libraries and time points. The identifier sequence was located at positions 91–98 relative to the Illumina primer and the barcode was located at positions 1–18. PCR products were purified two times over silica columns (Zymo Research) and quantified using the KAPA SYBR FAST qPCR Master Mix (Kapa Biosystems) on a Bio-Rad CFX machine. Samples were pooled and sequenced on an Illumina NextSeq instrument in single-end 100 bp mode.

## Analysis of bulk competition sequencing data

Illumina sequence reads were filtered for Phred scores > 20 and strict matching of the sequence to the expected template and identifier sequence. Reads that passed these filters were parsed based on the identifier sequence. For each condition/time-point identifier, each unique N18 read was counted. The unique N18 count file was then used to identify the frequency of each mutant using the variant-barcode association table. This barcoding strategy reduces the impact of bases misread by Illumina, as they result in barcodes that are not in our lookup table created by paired end sequencing and thus are discarded from the fitness analyses. To generate a cumulative count for each codon and amino acid variant in the library, the counts of each associated barcode were summed. To reduce experimental noise, selection coefficients were not calculated for variants with less than 100 reads at the 0 time point (*Boucher et al., 2014*). The average variant at the 0 time point had approximately 500 reads.

## Determination of selection coefficient

Selection coefficients were estimated using empiricIST (*Fragata et al., 2018*), a software package developed based on a previously published Markov Chain Monte Carlo (MCMC) approach (*Bank et al., 2014*). Briefly, we estimated individual growth rates and initial population sizes relative to the wild-type sequence simultaneously, based on a model of exponential growth and multinomial sampling of sequencing reads independently at each time point. For each mutant we obtained 10,000 posterior samples for the growth rate and initial population using a Metropolis-Hastings algorithm. The resulting growth rate estimates correspond to the median of 1000 samples of the posterior. Subsequently, selection coefficients (s) were scaled so that the average stop codon in each environmental condition represented a null allele (s = −1). For the second replicate in standard conditions, we noted a small fitness defect (s ≈ −0.2) for wild-type synonyms at positions 679–709 relative to other positions. We do not understand the source of this behavior, and chose to normalize to wild-type synonyms from 1 to 678 for this condition and to exclude positions 679–709 from analyses that include the second replicate of standard conditions. We did not observe this behavior in any other condition including the first standard condition replicate. Variants were categorized as having wild-type-like, beneficial, intermediate, or deleterious fitness based on the comparison of their selection coefficients with the distribution of wild-type synonyms and stop codons in each condition

(*Figure 2—figure supplement 7*) in the following manner; Wild-type-like: variants with selection coefficients within two standard deviations (SD) of the mean of wild-type synonyms; Beneficial: variants with selection coefficients above two SD of wild-type synonyms; Strongly deleterious: variants with selection coefficients within two SD of stop codons; Intermediate: variants with selection coefficients between those of stop-like and wild-type-like. Where stated, the average selection coefficient was calculated as the mean selection coefficient of all mutations at a position excluding that of the stop codon.

## Structural analysis

The solvent accessible surface area was computed by the algorithm of *Lee and Richards (1971)* using the PDB 2cg9 structure with the chains for Sba1 removed. The Blosom score was derived from the Blosom62 matrix (*Henikoff and Henikoff, 1992*). Evolutionary conservation was calculated with an alignment of homologs from diverse species using the ConSurf server (*Ashkenazy et al., 2016*). The change in protein stability upon mutation (ΔΔG) was predicted by the PoPMuSic server (*Dehouck et al., 2011*). Distance from the γ-phosphate of ATP to the C-α of each amino acid residue was calculated using Pymol. Physico-chemical properties of the amino acids were retrieved from *Abriata et al. (2015)*. Correlation coefficients were calculated by Pearson product-moment correlations unless otherwise stated.

## Random simulations to assess clustering of mutations

To assess if classes of mutations (e.g. temperature-sensitive mutations) clustered at positions more than expected based on chance, we compared the observed distribution of mutations to random simulations. For the random simulations, we randomly selected a position for the number of observed mutations and stored the clustering distribution (e.g. the number of positions with 0, 1, 2, 3, etc. simulated mutations). We performed 1000 simulations and used the average and standard deviation from these simulations to define statistical cutoffs for random expectations.

## Yeast growth analysis

Individual variants of Hsp90 were generated by site-directed mutagenesis and confirmed by Sanger sequencing. Variants were cloned in a p414 plasmid either under a low-expression, ADH promoter, or a high-expression, GPD promoter, as specified. Variants were generated by site directed mutagenesis and transformed into DBY288 cells. Selected transformed colonies were grown in liquid SRGal-W media to mid-log phase at 30°C, washed three times and grown in shutoff media (SD-W) for 10 hrs at 30°C, and then either kept at 30°C or shifted to 37°C as indicated. After sufficient time to stall the growth of control cells lacking a rescue copy of Hsp90 (~16 hr), cell density was monitored based on absorbance at 600 nm over time and fit to an exponential growth curve to quantify growth rate. Growth estimates were based on individual growth curves with at least four timepoints over an eight hour period. Using this approach we routinely observe measurement noise of 2–5%.

## Analysis of Hsp90 expression by western blot

To analyze expression levels of Hsp90, cells were grown for the specified time in SD-W or the indicated environmental condition. $10^8$ yeast cells were collected by centrifugation and frozen as pellets at −80°C. Cells were lysed by vortexing the thawed pellets with glass beads in lysis buffer (50 mM Tris-HCl pH 7.5, 5 mM EDTA and 10 mM PMSF), followed by addition of 2% Sodium dodecyl sulfate (SDS). Lysed cells were centrifuged at 18,000 g for 1 min to remove debris, and the protein concentration of the supernatants was determined using a BCA protein assay kit (Pierce) compared to a Bovine Serum Albumin (BSA) protein standard. 15 µg of total cellular protein was resolved by SDS-PAGE, transferred to a PVDF membrane, and Hsp90 was probed using an anti-human Hsp90 α/β antibody that cross reacts with yeast Hsp90 (Cayman chemical).

## Natural variation in Hsp90 sequence

We analyzed sequence variation in a previously described alignment of Hsp90 protein sequences from 261 eukaryotic species that broadly span a billion years of evolutionary distance (*Starr et al., 2018*).

## Acknowledgements

Thanks to Tyler Starr for providing the alignment of Hsp90 sequences used to assess natural variation. This work was supported by grants from the National Institutes of Health (R01-GM112844 to DNAB and F32-GM119205 to JMF). IF was supported by a postdoctoral fellowship from the FCT (Fundação para a Ciência e a Tecnologia) within the project JPIAMR/0001/2016. CB is grateful for support from EMBO Installation Grant IG4152 and ERC Starting Grant 804569 - FIT2GO.

## Additional information

### Funding

| Funder | Grant reference number | Author |
|---|---|---|
| National Institutes of Health | R01-GM112844 | Julia M Flynn<br>Ammeret Rossouw<br>Pamela Cote-Hammarlof<br>David Mavor<br>Carl Hollins III<br>Daniel NA Bolon |
| National Institutes of Health | F32-GM119205 | Julia M Flynn |
| Fundação para a Ciência e a Tecnologia | JPIAMR/0001/2016 | Inês Fragata |
| EMBO | EMBO Installation Grant IG4152 | Claudia Bank |
| European Research Council | ERC Starting Grant: 804569-FIT2GO | Claudia Bank |

The funders had no role in study design, data collection and interpretation, or the decision to submit the work for publication.

### Author contributions

Julia M Flynn, Conceptualization, Data curation, Software, Formal analysis, Validation, Investigation, Visualization, Methodology, Writing - original draft, Project administration, Writing - review and editing; Ammeret Rossouw, Data curation, Software, Investigation, Methodology; Pamela Cote-Hammarlof, Conceptualization, Investigation, Methodology, Writing - review and editing; Inês Fragata, Data curation, Software, Formal analysis, Validation, Writing - original draft, Writing - review and editing; David Mavor, Data curation, Software, Formal analysis; Carl Hollins III, Formal analysis, Validation, Writing - review and editing; Claudia Bank, Data curation, Software, Formal analysis, Funding acquisition, Validation, Writing - original draft, Writing - review and editing; Daniel NA Bolon, Conceptualization, Resources, Data curation, Software, Formal analysis, Supervision, Funding acquisition, Validation, Investigation, Visualization, Methodology, Writing - original draft, Project administration, Writing - review and editing

### Author ORCIDs

Julia M Flynn (iD) https://orcid.org/0000-0002-5490-393X
Inês Fragata (iD) https://orcid.org/0000-0001-6865-1510
Carl Hollins III (iD) https://orcid.org/0000-0003-0410-9639
Claudia Bank (iD) http://orcid.org/0000-0003-4730-758X
Daniel NA Bolon (iD) https://orcid.org/0000-0001-5857-6676

### Decision letter and Author response

Decision letter https://doi.org/10.7554/eLife.53810.sa1
Author response https://doi.org/10.7554/eLife.53810.sa2

# Additional files

## Supplementary files

- Supplementary file 1. List of oligomers used in this study.

- Transparent reporting form

## Data availability

Next generation sequencing data has been deposited to the NCBI short read archive (Project # PRJNA593726). Tabulated raw counts of all variants in all conditions are included in the manuscript in Figure 1—source data 1 and Figure 2—source data 2. Source data files have been provided for Figure 1, 2, 3, 4, 5 and 6.

The following dataset was generated:

| Author(s) | Year | Dataset title | Dataset URL | Database and Identifier |
|---|---|---|---|---|
| Flynn JM, Rossouw A, Cote-Hammerlof PA, Fragata I, Mavor D, HollinsIII C, Bank C, Bolon DNA | 2019 | Comprehensive fitness maps of Hsp90 show widespread environmental dependence | https://www.ncbi.nlm.nih.gov/bioproject/PRJNA593726/ | NCBI BioProject, PRJNA593726 |

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
