## [Decision Letter]

**Acceptance summary:**

The authors performed a mutational scanning experiment of Hsp90, a large heat-shock protein involved in regulating multiple cellular processes. They mutated all positions of this large protein to other amino acids and measured the growth of each mutant in standard condition and five stress conditions that may require Hsp90's function. The authors found that many variants are deleterious, as expected, and that these cluster in specific regions and tend not to be found in Hsp90 orthologs. The authors found that some mutations are beneficial, but these tend to be condition-specific, suggesting that evolution in fluctuating environments would favor overall robustness rather than allele variants that would be beneficial in specific conditions. This study reports one of the most extensive genotype-fitness maps of a protein performed to date and brings novel hypotheses regarding the evolution of proteins that have a large array of molecular functions and binding partners.

**Decision letter after peer review:**

Thank you for submitting your article "Comprehensive fitness maps of Hsp90 show widespread environmental dependence" for consideration by *eLife*. Your article has been reviewed by three peer reviewers, one of whom is a member of our Board of Reviewing Editors, and the evaluation has been overseen by Patricia Wittkopp as the Senior Editor. The reviewers have opted to remain anonymous.

The reviewers have discussed the reviews with one another and the Reviewing Editor has drafted this decision to help you prepare a revised submission.

Flynn et al., carried out a comprehensive deep mutational scanning of HSP90 in S cerevisiae (709 amino acids long). They measure the fitness effects of all possible single mutations and find that these effects are condition dependent. Their results allow to propose further models on the evolution of proteins such as HSP90 that play a role in stress response in variable environments. Overall, the study is elegantly designed, and the results are well presented. The three reviewers found the results highly interesting and found the scale of the experiments impressive. However, they raised several major points that need to be addressed. Some of them relate to the technical aspects of the work, others to the data analysis and finally, some relate to the potential broader impact of the results and interpretation.

The major comments are combined in a single list here:

1) The analysis of variability in the mutational sensitivity of synonymous mutations (discussed in subsection "Hsp90 potential for adaptation to environmental stress") seems a bit reductive. The analysis does not include the comparison of synonymous mutations at the beginning of the gene with the rest of synonymous mutations. The authors could have shown position-wise selection coefficients of synonymous mutations in a figure just to give a context of the data. In order to provide sense of the scale of variation, variability in synonymous mutations could be compared with that of non-synonymous mutations. Apart from this, there seems to be some inconsistent deleteriousness of some of the synonymous mutations (mentioned in subsection "Determination of Selection Coefficient"). I think it should be acknowledged in the results.

2) Figure 5—figure supplement 4B,C: statistical tests would be needed to draw a conclusion from the data.

3) Analysis of stop mutations

Similar to variability in the selection coefficients of the synonymous mutations, evidently there is also an extent of variability in the selection coefficients of stop mutations too (as seen in Figure 2—figure supplement 7). In my opinion, this aspect should have been addressed in the results. Also, are there any factors that determine the variability in the selection coefficients of stop mutations, apart from the position of the stop mutation?

4) Estimation of the level of between-replicate variation of deleterious mutations as a function of depth of sequencing. In the Results section, the authors acknowledge that the between-replicate variation is larger for the most deleterious mutations. As deleterious mutations are central to the results of this study, I think this point needs to be substantiated. In order to do so, an estimation of the level of between-replicate variation of deleterious mutations as a function of depth of sequencing could be shown.

5) Network analysis of altered protein-protein interactions

One of the most striking results of the study was that the beneficial mutations in elevated temperature condition and in case of diamide stress were found to be preferentially located on the binding surfaces of HSP90. However, the analysis to uncover the cause of such effect seems underdeveloped. Are there ways with which this could be considered further? Computationally, based on the mutationally altered binding between the protein interfaces, it could be possible to simulate the network rewiring and deduce the causes of the beneficial effect. Although not mandatory, such analyses would help in providing a network context to the study.

6) Relative strengths of the constraints. At various places in the text, the authors have analysed different constraints that guide mutational fitness effects while considering subsets of positions eg. ATP binding region, buried residues etc. However, I believe it could be helpful to analyse such strengths of constraints at the whole protein level. The estimation the relative strengths of the constraints and then ranking the constraints based on their strengths could be helpful in comparing the features of mutational sensitivity of HSP90 with other proteins such as with gentamicin-resistant gene in bacteria that they cite (Dandage et al., 2018). Such analysis could help in identifying the potentially unique features that are specific to HSP90.

7) Position-wise clustering of the variants

Figure 4B and Figure 5B: The analysis shown is difficult to understand. Either modifications in the plot or the caption is needed. The significance of the difference should be tested with a statistical test. How are the 'independent expectations' calculated?

8) The paralog of HSP90 is not discussed in the text. However, an earlier study showed that (https://journals.plos.org/plosbiology/article?id=10.1371/journal.pbio.1000347) one paralog is induced in expression when the other paralog is deleted (Figure 2, Hsp82). This seems to be a potential confounding factor if some mutations reduce protein abundance and in response the other copy's expression is induced. This may make some mutations be buffered by the other copy and not others. The strain used in this study appears to be the one described previously (Jiang et al., 2013) as a Hsc82 knockout. Therefore, interpretation of the fitness effects of Hsp82 variants is confounded by the absence of Hsc82. In nature, it is possible/likely that redundancy of the Hsp90 paralogs enables Hsp82 to navigate areas of the fitness landscape that appear selectively inaccessible in the present study (although there is now evidence that these paralogs exhibit some functional differences (https://www.ncbi.nlm.nih.gov/pmc/articles/PMC6689086/)). Relaxed constraint could on influence molecular evolution of Hsp90, specifically in regions that bind to clients and/or co-chaperones.

In addition, given the artificial context in which the protein is expressed, it seems likely that this system fails to recapitulate the native transcriptional, post-transcriptional, or translational regulation of the native system. While I do not doubt that the authors have devised an elegant system for assessing the effect of the various mutants on the Hsp90 protein (and in some cases the relevant mRNA, in the case of meaningfully important synonyms), it seems likely that the native gene(s) are under a much more complex selective pressure. I understand that fully addressing this concern experimentally is likely beyond the scope of the current study, but it needs to be considered in the Discussion section.

9) Another major point relates to expression levels (Hsp82 as stated in subsection "Impact of stress conditions on mutational sensitivity of Hsp90"). Was expression of Hsp82 affected by the stress conditions? The manuscript references a previous paper that describes a 10-fold reduction in Hsp82 expression, presumably in standard conditions. Is Hsp82 downregulated to a similar in the stress conditions? I would want to see Hsp82 expression levels before drawing any conclusions about Hsp82 function in the stress conditions. Another issue relates to the protein. What is the half-life of HSP90 and what happens when transcription is shut off?

10) Other technical aspects would be to be better described. For instance, the copy number and auxotrophy of the Hsp90 mutant expression plasmid are not indicated. Moreover, the genetic background of the experimental strains are also insufficiently described. Other questions also arise when trying to make sense of the experimental details. Which paralog is deleted and which is under the control of the galactose-inducible promoter? How many technical replicates per environment? I think duplicates for the standard condition and none for the other selective conditions? But this was hard to tell. How does noise affect the estimation of selective coefficients? Are the authors convinced that the behavior of this promoter is consistent across stress conditions? I would also like to know the copy number of plasmids used for cloning and expressing the libraries.

11) Another concern is regarding the scope and broad interest of the work described in the manuscript. The core experiment appears well designed and carefully conducted, but most of the key findings are intuitive or reflect existing knowledge of the Hsp90 protein or its intermolecular interactions. For reasons related to the three concerns described in more detail below, the strongest claims pertaining to evolution and selection are perhaps not best-addressed in this model system. Rather, it seems most suited for examining the behavior and function of the Hsp90 polypeptide in the abstract, and the findings here may be of interest to a more focused audience. Finally, there is limited validation, confirmation, or exploration of the findings from the selection experiment in orthogonal experimental systems.

12) One of the major novelties of the work is the systematic assessment of GXE. Since only a few studies have performed such comprehensive assays, it would be important to mention them and contrast them with the current work. I missed a comparison and discussion, or even just a mention, of the fitness landscapes of 23,000 tRNA genotypes across four different selective environments (Li and Zhang, 2018). In that paper, whereas the GxE was pervasive, the patterns detected were so simple that the fitness landscape in a given environment could be easily predicted from the fitness of a few genotypes in another environment using a piecewise linear regression model. Would the same approach work here? If that approach does not work with the Hsp90 data, could that indicate a fundamental difference in fitness landscapes and GxE interactions between RNA molecules and proteins?

13) My final significant question relates to the observation of minimal antagonistic pleiotropy (and little deleterious variation) amongst naturally occurring variants across different experimental conditions. The authors claim that this constitutes evidence for selection on "robustness" across fluctuating environments. I would propose that, rather, it may suggest that the molecular function (potentially modulo selective client binding, as the authors observe) is simply similar across environments and thus, "robustness" merely reflects that Hsp90 foldase function is an important cellular process (as reflected by the inviability of the HSP82 HSC82 double-deletion strain). Indeed, it seems most "essential" genes would likely exhibit this behavior by dint of the importance of maintaining their function. Perhaps this is a semantic difference; in any case the authors would do well to clarify this line of argument in the text.

[Editors' note: further revisions were suggested prior to acceptance, as described below.]

Thank you for resubmitting your work entitled "Comprehensive fitness maps of Hsp90 show widespread environmental dependence" for further consideration by *eLife*. Your revised article has been evaluated by Patricia Wittkopp (Senior Editor) and a Reviewing Editor.

The manuscript has been improved but there are some remaining issues that need to be addressed before acceptance, as outlined below:

1) The randomizations discussed in subsection "Constraint of mutational sensitivity at high temperature" should be described in the Materials and methods section.

2) The description of the different types of mutations in subsection "Constraint of mutational sensitivity at high temperature" could be better explained. Deleterious mutations are not always purged so it would be better to say "strongly deleterious mutations that are purged" to contrast with slightly deleterious ones (nearly-neutral).

3) The Western blot shown in Figure 6—figure supplement 1 does not include a loading control to show that the same amount of protein was loaded. If the amount of protein loaded was measured and normalized, please specify how otherwise it would be important to mention how you can compare the samples without appropriate controls.

4) It would be better to show the data mentioned here: "cells containing a null-rescue plasmid had stopped growing and Hsp90 was undetectable by Western blot (data not shown)."

5) The tabulated data used to generate figures is provided but the code used to generate this data is not provided. Please provide the code or make it available through an online public repositories such as GitHub or others.

---

## [Author Response]

Flynn et al., carried out a comprehensive deep mutational scanning of HSP90 in S cerevisiae (709 amino acids long). They measure the fitness effects of all possible single mutations and find that these effects are condition dependent. Their results allow to propose further models on the evolution of proteins such as HSP90 that play a role in stress response in variable environments. Overall, the study is elegantly designed, and the results are well presented. The three reviewers found the results highly interesting and found the scale of the experiments impressive. However, they raised several major points that need to be addressed. Some of them relate to the technical aspects of the work, others to the data analysis and finally, some relate to the potential broader impact of the results and interpretation.The major comments are combined in a single list here:1) The analysis of variability in the mutational sensitivity of synonymous mutations (discussed in subsection "Hsp90 potential for adaptation to environmental stress") seems a bit reductive. The analysis does not include the comparison of synonymous mutations at the beginning of the gene with the rest of synonymous mutations. The authors could have shown position-wise selection coefficients of synonymous mutations in a figure just to give a context of the data. In order to provide sense of the scale of variation, variability in synonymous mutations could be compared with that of non-synonymous mutations. Apart from this, there seems to be some inconsistent deleteriousness of some of the synonymous mutations (mentioned in subsection "Determination of Selection Coefficient"). I think it should be acknowledged in the results.

We have added several supplementary figures that are outlined below in order to provide further analyses of synonymous substitutions:

Figure 2—figure supplement 8B has been added to show position-wise selection coefficients of wild-type synonyms for all conditions, indicating the variability between conditions

Figure 5—figure supplement 4A has been added to show the position-wide variability in selection coefficients for synonymous codons compared to the position-wide average selection coefficients of all mutations to emphasize that the high variability for synonymous mutations at the N-terminus coincides with a high number of other beneficial mutations that we speculate may be linked to nucleotide dependence in this region.

To provide a sense of scale of variation, Figure 2—figure supplement 6 has been added to show variability in synonymous mutations compared to non-synonymous in all conditions.

Figure 1—figure supplement 1 has been added to show position-wise selection coefficients of synonymous mutations for standard replicate 1 and replicate 2. This highlights the deleteriousness of the synonymous mutations at the C terminus in the second replicate.

This has also been acknowledged in the Results section:

"For the second replicate we noted a small fitness defect (s~0.05) for wildtype synonyms at positions 679-709 relative to other positions (Figure 1—figure supplement 1). We did not see this behavior in any other condition or replicate tested and do not understand its source."

2) Figure 5—figure supplement 4B,C: statistical tests would be needed to draw a conclusion from the data.

Figure 5—figure supplement 4A has been added to show the position-wide variability in selection coefficients for synonymous codons compared to the position-wide average selection coefficient of all mutations. Synonymous mutations that were deemed beneficial or deleterious after a Bonferroni correction are noted with asterisks. The strongest beneficial mutation is located at position 2 and we confirmed that this mutation confers a growth advantage when analyzed in isolation and that it increases the amount of Hsp90 protein expressed in cells as shown in Figure 5—figure supplement 4.

3) Analysis of stop mutationsSimilar to variability in the selection coefficients of the synonymous mutations, evidently there is also an extent of variability in the selection coefficients of stop mutations too (as seen in Figure 2—figure supplement 7). In my opinion, this aspect should have been addressed in the results. Also, are there any factors that determine the variability in the selection coefficients of stop mutations, apart from the position of the stop mutation?4) Estimation of the level of between-replicate variation of deleterious mutations as a function of depth of sequencing. In the Results section, the authors acknowledge that the between-replicate variation is larger for the most deleterious mutations. As deleterious mutations are central to the results of this study, I think this point needs to be substantiated. In order to do so, an estimation of the level of between-replicate variation of deleterious mutations as a function of depth of sequencing could be shown.

The following figures have been added to address comments 3 and 4:

Figure 1—figure supplement 1 has been added to show the position-wise variability in selection coefficients for stops in standard replicate 1 compared to standard replicate 2. This shows a position-independent variability across the gene except for the last 32 residues that are dispensable for function.

Figure 2—figure supplement 6 has been added to show the distribution of stops compared to wildtype synonyms in all conditions.

Figure 1—figure supplement 3 has been added to show an analysis of underlying reasons for noise in stops. Part A shows that stops are partially depleted by the first time point. Part B shows an analysis of the between replicate variation of stops as a function of sequencing depth showing that variation increases with decreased reads. Part C shows that the distribution in stop selection coefficients is independent of the stop codon (e.g. TGA, TGG, TAG).

The following has been added in the Results section to address this variability:

"Of note, variants with strongly deleterious effects exhibited the greatest variation between replicates, consistent with the noise inherent in estimating the frequency of rapidly depleting variants. The stop codons were already partially depleted from the cells at the 0 time point, likely contributing to their variation between replicates (Figure 1—figure supplement 3A). In accordance with this, there is a higher variation in selection coefficients between replicates for stop codons with the lowest initial reads (Figure 1—figure supplement 3B). Stop codon fitness is similar for all three stop codons (Figure 1—figure supplement 3C) and at positions across Hsp90 with exception of the last 32 positions that have previously been shown to be dispensable for its viability (Louvion et al., 1996) (Figure 1—figure supplement 1)."

5) Network analysis of altered protein-protein interactionsOne of the most striking results of the study was that the beneficial mutations in elevated temperature condition and in case of diamide stress were found to be preferentially located on the binding surfaces of HSP90. However, the analysis to uncover the cause of such effect seems underdeveloped. Are there ways with which this could be considered further? Computationally, based on the mutationally altered binding between the protein interfaces, it could be possible to simulate the network rewiring and deduce the causes of the beneficial effect. Although not mandatory, such analyses would help in providing a network context to the study.

Discussion of this has been added in the Discussion section:

"In contrast, beneficial mutations tended to be on the surface of Hsp90 and at contact sites with binding partners, suggesting that change-of-function mutations may be predominantly governed by alterations to binding interactions. Mutations that disrupt binding to certain clients can lead to the re-prioritization of others, which, due to Hsp90's central role in numerous cellular pathways, has the potential to modify integral networks in response to stress. Once more comprehensive data is available on Hsp90-client binding sites, it may be possible to simulate this rewiring of cellular networks, providing insight into the causes of the beneficial mutations. However, presently, the large number of clients with unknown binding sites makes these analyses challenging. In the future, comparing Hsp90 client-interactomes (Taipale et al., 2012) may help delineate adaptive biochemical mechanisms."

6) Relative strengths of the constraints. At various places in the text, the authors have analysed different constraints that guide mutational fitness effects while considering subsets of positions eg. ATP binding region, buried residues etc. However, I believe it could be helpful to analyse such strengths of constraints at the whole protein level. The estimation the relative strengths of the constraints and then ranking the constraints based on their strengths could be helpful in comparing the features of mutational sensitivity of HSP90 with other proteins such as with gentamicin-resistant gene in bacteria that they cite (Dandage et al., 2018). Such analysis could help in identifying the potentially unique features that are specific to HSP90.

Figure 6—figure supplement 2 and Figure 6—figure supplement 2, source data has been added to analyze the strength of constraints at a whole protein level in a similar manner as Dandage et al., 2018.

Subsection "Relationship to prior work" the following has been added to discuss this analysis, compare the features with that of Gentamicin-resistant gene, and point out unique features of Hsp90:

"A handful of studies have assessed the impact of environment on the fitness landscape of genes Hietpas, 2013(Dandage et al., 2018; ; Li and Zhang, 2018; Mavor et al., 2016). For example, Dandage, Chakraborty and colleges investigated the effects of temperature and chemical chaperones on the fitness landscape of the Gentamicin resistance gene and found that protein stability and distance to the ligand binding site are the molecular properties with the strongest correlations with fitness (Dandage et al., 2018). To understand the strength of the molecular constraints on Hsp90 on a whole protein level, we performed similar analyses (Figure 6—figure supplement 2). Consistent with the Gentamicin study, we find the features that best correlate with fitness are protein stability and distance from the active site. The constraint of protein stability is the highest at 37°C, indicating increased dependence on stability at the higher temperature. In addition, distance from the ATP binding site imposes strong molecular constraints on Hsp90, signifying the importance of ATP hydrolysis on Hsp90 function. While individual features correlate with fitness effects and show environmental dependence, single features are unable to capture the majority of observed variance in fitness effects, consistent with a complex set of physical properties that underlie fitness effects in both proteins."

7) Position-wise clustering of the variantsFigure 4B and Figure 5B: The analysis shown is difficult to understand. Either modifications in the plot or the caption is needed. The significance of the difference should be tested with a statistical test. How are the 'independent expectations' calculated?

We have reworded the figure legend and the Results section to clarify descriptions of these figures:

"We found that *ts* mutations tended to concentrate at certain amino acid positions of Hsp90 (Figure 4B). The clustering of *ts* mutations was significant compared to random simulations. Positions with greater than four *ts* mutations were spread across all three domains of Hsp90 (Figure 4C) with the largest cluster occurring in the C domain of Hsp90."

We also added a statistical test of the significance of the clustered mutants. The figure legends have been updated to clarify Figure 4B and Figure 5B.

8) The paralog of HSP90 is not discussed in the text. However, an earlier study showed that (https://journals.plos.org/plosbiology/article?id=10.1371/journal.pbio.1000347) one paralog is induced in expression when the other paralog is deleted (Figure 2, Hsp82). This seems to be a potential confounding factor if some mutations reduce protein abundance and in response the other copy's expression is induced. This may make some mutations be buffered by the other copy and not others. The strain used in this study appears to be the one described previously (Jiang et al., 2013) as a Hsc82 knockout. Therefore, interpretation of the fitness effects of Hsp82 variants is confounded by the absence of Hsc82. In nature, it is possible/likely that redundancy of the Hsp90 paralogs enables Hsp82 to navigate areas of the fitness landscape that appear selectively inaccessible in the present study (although there is now evidence that these paralogs exhibit some functional differences (https://www.ncbi.nlm.nih.gov/pmc/articles/PMC6689086/)). Relaxed constraint could on influence molecular evolution of Hsp90, specifically in regions that bind to clients and/or co-chaperones.

In addition, given the artificial context in which the protein is expressed, it seems likely that this system fails to recapitulate the native transcriptional, post-transcriptional, or translational regulation of the native system. While I do not doubt that the authors have devised an elegant system for assessing the effect of the various mutants on the Hsp90 protein (and in some cases the relevant mRNA, in the case of meaningfully important synonyms), it seems likely that the native gene(s) are under a much more complex selective pressure. I understand that fully addressing this concern experimentally is likely beyond the scope of the current study, but it needs to be considered in the Discussion section.

We have addressed both of these points to the Discussion section.

"Our experimental setup has limitations that we have tried to account for in our analyses and conclusions. For example, we measured the fitness effects of Hsp90 under artificially low expression where yeast growth rates are tightly coupled to function of Hsp90 in order to provide a sensitive readout of fitness defects (Jiang et al., 2013). Expression of Hsp90 under this promoter remains stable in the stresses tested (Figure 6—figure supplement 1). However, this defined promoter does not capture the native transcriptional regulation and may not fully recapitulate translational and post-translational regulation controlling *hsp82*. While these levels of regulation of Hsp90 are clearly important physiologically, the sensitive readouts of fitness that we measured appear to capture critical features of Hsp90 with regards to biochemical function and evolutionary mechanism. For example, virtually all deleterious mutants measured in this study under stress conditions appear to have also been subject to purifying selection in nature.

In addition, the experimental strain used in this study is deleted for the constitutively expressed paralog of *hsp82*, *hsc82*. *Hsp82* and *hsc82* are functionally overlapping, essential genes with 97% sequence identify (16 amino acid differences) that can compensate for each other's loss-of-function in normal growth conditions (Girstmair et al., 2019). The high sequence identity between the two paralogs indicates that they are both under similar selection pressure. Despite the high sequence identity, a number of distinct differences have been noted in stability, conformational cycles, and client interactomes. Experimental evidence indicates that *hsp82* is more stress-specific, and more stable to unfolding (Girstmair et al., 2019). Further efforts will be required to resolve how distinctions between Hsp90 paralogs contribute to function and selection."

9) Another major point relates to expression levels (Hsp82 as stated in subsection "Impact of stress conditions on mutational sensitivity of Hsp90"). Was expression of Hsp82 affected by the stress conditions? The manuscript references a previous paper that describes a 10-fold reduction in Hsp82 expression, presumably in standard conditions. Is Hsp82 downregulated to a similar in the stress conditions? I would want to see Hsp82 expression levels before drawing any conclusions about Hsp82 function in the stress conditions. Another issue relates to the protein. What is the half-life of HSP90 and what happens when transcription is shut off?

The Hsp90 library is expressed under a constitutive low-expression level promoter (approximately 10-fold reduction in expression) and expression levels remain generally unchanged by the various environmental conditions. Normally in the cell, *hsp82* expression would be upregulated up to 10-fold in all stress conditions tested with the exception of salt (this is explained in the result section).

The first two paragraphs of the Results section have been updated to clarify the experimental setup.

Figure 6—figure supplement 1 has been added to the Results section to show that the expression of Hsp90 under this promoter remains stable in the environmental conditions tested.

The following has been added in the Discussion section to discuss the role expression levels play in our experimental setup:

"For example, we measured the fitness effects of Hsp90 under artificially low expression where yeast growth rates are tightly coupled to function of Hsp90 in order to provide a sensitive readout of fitness defects (Jiang et al., 2013). Expression of Hsp90 under this promoter remains stable in the stresses tested (Figure 6—figure supplement 1). However, this defined promoter does not capture the native transcriptional regulation and may not fully recapitulate translational and post-translational regulation controlling *hsp82*. While these levels of regulation of Hsp90 are clearly important physiologically, the sensitive readouts of fitness that we measured appear to capture critical features of Hsp90 with regards to biochemical function and evolutionary mechanism. For example, virtually all deleterious mutants measured in this study under stress conditions appear to have also been subject to purifying selection in nature."

We have not measured the half-life of Hsp90, however, cells that contain a null-rescue plasmid instead of a plasmid expressing Hsp90 stop growing after 10 hours after the shift from galactose to dextrose and Hsp90 protein is undetectable by western blot at this time point. We added text to describe this in the Materials and methods section:

"Thawed cells were amplified in SRGal-W for 24 hours, and then shifted to shutoff conditions by centrifugation, washing, and resuspension in 300 mL of synthetic dextrose lacking tryptophan (SD-W) for 12 hours at 30Â°C. At this time, cells containing a null-rescue plasmid had stopped growing and Hsp90 was undetectable by Western blot (data not shown)."

10) Other technical aspects would be to be better described. For instance, the copy number and auxotrophy of the Hsp90 mutant expression plasmid are not indicated. Moreover, the genetic background of the experimental strains are also insufficiently described. Other questions also arise when trying to make sense of the experimental details. Which paralog is deleted and which is under the control of the galactose-inducible promoter? How many technical replicates per environment? I think duplicates for the standard condition and none for the other selective conditions? But this was hard to tell. How does noise affect the estimation of selective coefficients? Are the authors convinced that the behavior of this promoter is consistent across stress conditions? I would also like to know the copy number of plasmids used for cloning and expressing the libraries.

The copy number, auxotrophy of the plasmid, and genetic background of the strain have been described in more detail in the Materials and methods section.

Experimental details have been clarified in the second paragraph of the Results section and in the Materials and methods section.

There is a technical replicate of the standard conditions but not the other conditions.

We assessed measurement noise by comparing replicates in the standard conditions (see Figure 1B).

Figure 6—figure supplement 1 has been added to show that the expression levels of Hsp90 are consistent under the ADH promoter in the different environmental conditions.11) Another concern is regarding the scope and broad interest of the work described in the manuscript. The core experiment appears well designed and carefully conducted, but most of the key findings are intuitive or reflect existing knowledge of the Hsp90 protein or its intermolecular interactions. For reasons related to the three concerns described in more detail below, the strongest claims pertaining to evolution and selection are perhaps not best-addressed in this model system. Rather, it seems most suited for examining the behavior and function of the Hsp90 polypeptide in the abstract, and the findings here may be of interest to a more focused audience. Finally, there is limited validation, confirmation, or exploration of the findings from the selection experiment in orthogonal experimental systems.

We have performed global analyses of the biophysical features underlying fitness effects and compared our findings to other environmental studies of protein fitness landscapes. We have compared the fitness effects in the N domain to an independent study, and our key conclusions are all supported by multiple observations with appropriate statistical tests. We agree that high throughput studies can be challenging to validate, but we have tried to account for this in our approach and discussion.12) One of the major novelties of the work is the systematic assessment of GXE. Since only a few studies have performed such comprehensive assays, it would be important to mention them and contrast them with the current work. I missed a comparison and discussion, or even just a mention, of the fitness landscapes of 23,000 tRNA genotypes across four different selective environments (Li and Zhang, 2018). In that paper, whereas the GxE was pervasive, the patterns detected were so simple that the fitness landscape in a given environment could be easily predicted from the fitness of a few genotypes in another environment using a piecewise linear regression model. Would the same approach work here? If that approach does not work with the Hsp90 data, could that indicate a fundamental difference in fitness landscapes and GxE interactions between RNA molecules and proteins?

A comparison of this study to other comprehensive GxE assays has been added in the Discussion section "Relationship to prior work".

A comparison and discussion to the Li and Zhang paper has been added to the Discussion section:

"In another study of the effect of environment on mutational fitness, Li and Zhang detected pervasive genotype-by-environment interactions between a yeast tRNA gene and environment (Li and Zhang, 2018). They found that the correlation of the fitness between mutations in each tested environment was linear such that the fitness landscape in one environment together with a change in slope could be used to accurately predict fitness effects in the second environment. In this study, we observed a large impact of environment on Hsp90 fitness, however, we observe many fitness effects that deviate from a linear relationship between environments. While linear models can predict the fitness of some mutations in different environments, it would not predict many of the types of mutations that are focuses of this study, such as mutations that exhibit an adaptive trade-off, those with beneficial effects in one environment that become deleterious in another. In addition, the linear model would not predict the large group of *ts* mutations with wildtype fitness in standard conditions and null fitness at 37°C. As environmental-dependent protein fitness landscapes are analyzed for an increasing set of genes, it will provide opportunities to explore how different protein properties such as the number of binding partners may contribute to global trends."

13) My final significant question relates to the observation of minimal antagonistic pleiotropy (and little deleterious variation) amongst naturally occurring variants across different experimental conditions. The authors claim that this constitutes evidence for selection on "robustness" across fluctuating environments. I would propose that, rather, it may suggest that the molecular function (potentially modulo selective client binding, as the authors observe) is simply similar across environments and thus, "robustness" merely reflects that Hsp90 foldase function is an important cellular process (as reflected by the inviability of the HSP82 HSC82 double-deletion strain). Indeed, it seems most "essential" genes would likely exhibit this behavior by dint of the importance of maintaining their function. Perhaps this is a semantic difference; in any case the authors would do well to clarify this line of argument in the text.

This point has been clarified in the text (subsection "Natural selection favors Hsp90 variants that are robust to environment"):

"Whereas naturally occurring amino acids in Hsp90 were rarely deleterious in any experimental condition, they were similarly likely to provide a growth benefit compared to all possible amino acids (5%). This observation indicates that condition-dependent fitness benefits are not a major determinant of natural variation in Hsp90 sequences. Instead our results indicate that natural selection has favored Hsp90 substitutions that are robust to multiple stressful conditions (Figure 6D). Beneficial mutations in heat and diamide indicate that there is room for improvement in Hsp90 function in individual conditions. The clustering of beneficial mutations at known binding interfaces suggests that the optimal binding affinity for partner proteins may depend on growth conditions. We propose that natural variants of Hsp90 have been selected for binding properties that are robust to different stresses rather than specific to individual conditions."

[Editors' note: further revisions were suggested prior to acceptance, as described below.]Thank you for resubmitting your work entitled "Comprehensive fitness maps of Hsp90 show widespread environmental dependence" for further consideration by eLife. Your revised article has been evaluated by Patricia Wittkopp (Senior Editor) and a Reviewing Editor.The manuscript has been improved but there are some remaining issues that need to be addressed before acceptance, as outlined below:1) The randomizations discussed in subsection "Constraint of mutational sensitivity at high temperature" should be described in the Materials and methods section.

We have added subsection "Random simulations to assess clustering of mutations" to discuss the randomizations:

"Random simulations to assess clustering of mutations

To assess if classes of mutations (e.g. temperature sensitive mutations) clustered at positions more than expected based on chance, we compared the observed distribution of mutations to random simulations. For the random simulations, we randomly selected a position for the number of observed mutations and stored the clustering distribution (for example the number of positions with 0, 1, 2, 3, etc. simulated mutations). We performed 1000 simulations and used the average and standard deviation from these simulations to define statistical cutoffs for random expectations."

2) The description of the different types of mutations in subsection "Constraint of mutational sensitivity at high temperature" could be better explained. Deleterious mutations are not always purged so it would be better to say "strongly deleterious mutations that are purged" to contrast with slightly deleterious ones (nearly-neutral).

This sentence has been modified as follows:

"Depending on environmental conditions, mutations can be categorized into three classes: strongly deleterious mutations that are purged from populations by purifying selection, nearly-neutral mutations that are governed by stochastic processes, and beneficial mutations that tend to provide a selective advantage (Ohta, 1973)."

3) The Western blot shown in Figure 6—igure supplement 1 does not include a loading control to show that the same amount of protein was loaded. If the amount of protein loaded was measured and normalized, please specify how otherwise it would be important to mention how you can compare the samples without appropriate controls.

Subsection "Analysis of Hsp90 expression by Western" has been to describe the method used for the Western blots, including quantification of total protein concentration loaded:

"Analysis of Hsp90 expression by Western

To analyze expression levels of Hsp90, cells were grown for the specified time in SD-W or the indicated environmental condition. 10^8^ yeast cells were collected by centrifugation and frozen as pellets at -80°C. Cells were lysed by vortexing the thawed pellets with glass beads in lysis buffer (50 mM Tris-HCl pH 7.5, 5 mM EDTA and 10 mM PMSF), followed by addition of 2% Sodium dodecyl sulfate (SDS). Lysed cells were centrifuged at 18,000 g for 1 minute to remove debris, and the protein concentration of the supernatants was determined using a BCA protein assay kit (Pierce) compared to a Bovine Serum Albumin (BSA) protein standard. 15 ug of total cellular protein was resolved by SDS-PAGE, transferred to a PVDF membrane, and Hsp90 was probed using an anti-human Hsp90 Î±/Î² antibody that cross reacts with yeast Hsp90 (Cayman chemical)."

4) It would be better to show the data mentioned here: "cells containing a null-rescue plasmid had stopped growing and Hsp90 was undetectable by Western blot (data not shown)."

Figure 6—figure supplement 3 has been added to show that the cells had stopped growing and that Hsp90 was undetectable by Western blot after 12 hours of growth in dextrose.

5) The tabulated data used to generate figures is provided but the code used to generate this data is not provided. Please provide the code or make it available through an online public repositories such as GitHub or others.

The scripts used to generate the data (including both those for association of open read frame mutations with barcodes and tabulation of Hsp90 allele counts) have been added to Github at the following repositories:

https://github.com/JuliaFlynn/Barcode_ORF_assembly

https://github.com/JuliaFlynn/Tabulate_counts

Links to these repositories have been added to the Key Resource Table.